# Direct conversion of methane to aromatics and hydrogen via a heterogeneous trimetallic synergistic catalyst

Pengxi Zhu[1,2], Wenjuan Bian[1], Bin Liu[1,3], Hao Deng[1,3], Lucun Wang[1], Xiaozhou Huang[2], Stephanie L. Spence[4], Feng Lin[4], Chuancheng Duan[3], Dong Ding[1] ✉, Pei Dong[2] ✉ & Hanping Ding[1,5] ✉

Non-oxidative methane dehydro-aromatization reaction can co-produce hydrogen and benzene effectively on a molybdenum-zeolite based thermo-chemical catalyst, which is a very promising approach for natural-gas upgrading. However, the low methane conversion and aromatics selectivity and weak durability restrain the realistic application for industry. Here, a mechanism for enhancing catalysis activity on methane activation and carbon-carbon bond coupling has been found to promote conversion and selectivity simultaneously by adding platinum–bismuth alloy cluster to form a trimetallic catalyst on zeolite (Pt-Bi/Mo/ZSM-5). This bimetallic alloy cluster has synergistic interaction with molybdenum: the formed $CH_3^*$ from $Mo_2C$ on the external surface of zeolite can efficiently move on for C-C coupling on the surface of Pt-Bi particle to produce $C_2$ compounds, which are the key intermediates of oligomerization. This pathway is parallel with the catalysis on Mo inside the cage. This catalyst demonstrated 18.7% methane conversion and 69.4% benzene selectivity at 710 °C. With 95% methane/5% nitrogen feedstock, it exhibited robust stability with slow deactivation rate of 9.3% after 2 h and instant recovery of 98.6% activity after regeneration in hydrogen. The enhanced catalytic activity is strongly associated with synergistic interaction with Mo and ligand effects of alloys by extensive mechanism studies and DFT calculation.

Aromatics, in particular, BTX aromatics (benzene, toluene, and xylenes) are the essential building blocks for a wide range of consumer products such as adhesives, paints, packaging, and clothing. To date, the majority (~97%) of aromatics production relies on crude oil in the first place. The current state-of-art aromatics production mainly hinges on three approaches: steam cracking of naphtha, catalytic reforming, and coke-oven light oil[1–7], which require significant amounts of synthesis gas, in turn creating extensive emissions of carbon dioxide ($CO_2$) and increasing complexity of chemical process. Instead, the non-oxidative deprotonation of methane, also defined as methane dehydro-aromatization (MDA) reaction, is a chemical process to directly convert natural gas to aromatics and hydrogen as a co-product, which has been widely studied for decarbonizing the chemical production industry.

As a thermal catalytic process, the aromatics formation was reported by Wang et al. for the first time over a molybdenum on zeolite

[1]Energy and Environment Science & Technology, Idaho National Laboratory, Idaho Falls, ID 83415, USA. [2]Department of Mechanical Engineering, George Mason University, Fairfax, VA 22030, USA. [3]Department of Chemical Engineering, Kansas State University, Manhattan, KS 66506, USA. [4]Department of Chemistry, Virginia Tech, Blacksburg, VA 24061, USA. [5]School of Aerospace and Mechanical Engineering, University of Oklahoma, Norman, OK 73019, USA. ✉e-mail: dong.ding@inl.gov; pdong3@gmu.edu; hding@ou.edu

catalyst with promising conversion and selectivity, which opens the paradigm of monometallic catalyst for direct methane conversion[8]. Meanwhile, high selectivity to light aromatics depends on zeolite shape selectivity. A combination of $Mo_2C$ with ZSM-5 exhibits unique performance for the aromatization of lower alkanes[9,10]. This catalyst family has demonstrated high flexibility in adjusting composition and performance[11], e.g., Mo loading (2–6 wt%), Si/Al ratios (10–25) and temperature (700–800 °C), methane conversion from 7% to 17% and benzene selectivity from 50% to 80%[8,12–15]. However, the challenge remains mainly on the limited thermodynamics for higher conversion and serious coking, which is a ubiquitous obstacle for the application of Mo/zeolite catalysts.

From the aromatization mechanism perspective, the MDA reaction goes through three transient stages: activation/induction, formation of carbon double bond, and aromatization[16], as illustrated in the conventional pathway via A–C–D in Fig. 1. In this process, dehydrogenation and oligomerization of methane occur on active Mo sites to form $C_2$ intermediates such as ethene and acetylene, followed by cyclization producing aromatics and naphthalene on the Brønsted acid sites (BASs) in zeolite[17]. It is generally accepted that the partially reduced and/or carburized Mo-oxo species in zeolite channels such as $Mo_xC_y$ and oxycarbide ($MoO_xC_y$) serve as the active sites while the Mo carbides on the external surface are less active[18], and the proximity of the acidic proton-Mo sites is shown to be a key factor in determining the catalyst lifetime as the migration of Mo species to the zeolite external surface leads to quick catalyst deactivation[17]. In addition, coke may occur during fast kinetics. The catalytic pyrolysis of $CH_4$ on the active $Mo_2C/MoO_xC_y$ sites leads to amorphous coke deposits (soft coke), and oligomerization and/or cracking of the intermediates ($C_2H_4$), and polycondensation of formed aromatics on the BASs promotes polyaromatic carbonaceous deposits (hard coke)[19–21].

As $C_2$ species formation and available BAS/Brønsted acidic protons can largely determine benzene yielding, the design of catalyst and the morphology of catalytic metals on zeolite become essential factors. A recent finding shows that a Pt-Bi bimetallic catalyst can deliver $C_2$ selectivity >90% at moderate temperatures (600–700 °C) from non-oxidative methane coupling[22]. The high selectivity was attributed to the proposed hypothesis that surface Pt functions as an active site for methane activation while Bi promotes $C_2$ species formation and catalyst stability. From this point, forming a multifunctional Mo-X (X = metal) catalyst system could potentially combine the advantages from each metal to obtain desirable properties.

In this work, we report a trimetallic platinum–bismuth/molybdenum on zeolite structure (Pt-Bi/Mo-ZSM5) as a highly efficient thermochemical catalyst for co-synthesis of aromatics and hydrogen from methane. As demonstrated, the bimetallic Pt-Bi alloy facilitates methane conversion by 25% comparing to the conventional Mo-based catalysts by yielding a higher absolute selectivity towards benzene under high $CH_4$ conversion (e.g., ~70% at conversion of $CH_4$ = 18% under space velocity 1272 $mL \cdot g_{cat}^{-1} \cdot h^{-1}$). It is proposed that the synergistic interaction between bimetallic Pt-Bi alloy and Mo species creates a (A–B–D) pathway for elementary reaction on zeolite external surface, as shown in Fig. 1, which is parallel with the conventional (A–C–D) pathway via the acidic proton-Mo-C sites in the interior surface of zeolite cages. On the external surface, the activation of the C-H bond occurs on the $Mo_2C$, leading to dissociative adsorption of methane to form surface alkyl intermediate ($CH_3^*$). Instantly, the $CH_3^*$ is transferred to the surface of Pt-Bi particles for subsequential scission of C-H bonds to form $CH_2^*$ and then preferential C-C coupling of two $CH_2$ species instead of the deep dehydrogenation to improve the selectivity for $C_2$ products and the catalyst lifetime. On the other hand, the Pt-Bi alloy mitigates the mobility of the Mo species, possibly enhancing the retention of the Mo species inside the zeolite channels during reaction. To understand the synergy between Pt-Bi alloy and Mo species, the synthesis, microstructural analysis, performance evaluation, and mechanism exploration have been performed to elucidate the rational principle of how Pt-Bi bimetallic alloys promote methane activation and aromatization pathways and thus to format a thermochemical catalyst with good activity and stability for MDA application.

## Results

### Synthesis and characterization of the trimetallic Pt-Bi/Mo/ZSM-5 catalyst

To observe the morphology of the synthesized trimetallic catalyst, the high-angle annular dark-field scanning transmission electron microscopy (HAADF-STEM) imaging was shown in Fig. 2A–C to illustrate the

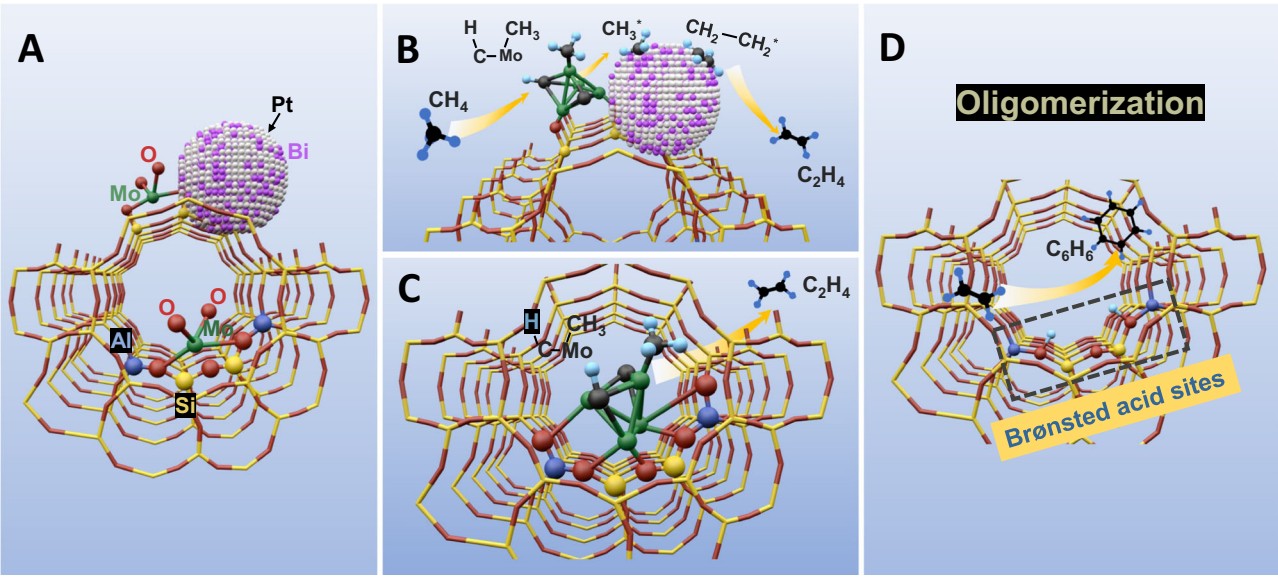

**Fig. 1 | A proposed elementary reaction pathway for activating C-H bond and coupling of C-C bond in MDA reaction.** A–C–D route (in the interior surface of zeolite cage): The reactions occur in the pores of zeolite where Mo species are located. The methane dehydrogenation and $C_2$ coupling are accomplished via molybdenum, and the subsequent oligomerization of $C_2$ intermediates is completed by BASs. A–B–D route (on zeolite external surface): The activation of C-H bond occurs on the $Mo_2C$ at surface to form $CH_3^*$ which transfers to the Pt-Bi alloy for C-C coupling and then for oligomerization reaction on BASs. Color scheme: Mo (green), C (black), Bi (purple), Si (yellow), Al (Dark blue), H (light blue), O (red) and Pt (white).

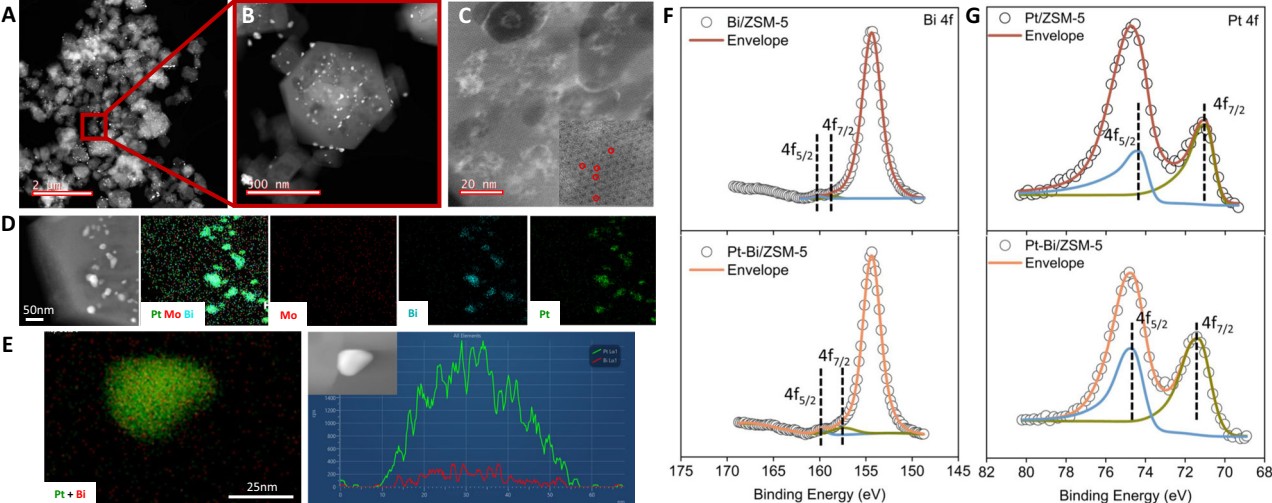

**Fig. 2 | Surface morphology and oxidation state analysis of as-prepared Pt-Bi/Mo/ZSM-5 catalysts. A–C** HAADF-STEM images of Pt-Bi/Mo/ZSM-5 surface: the circles in (**C**) indicate the in-channel Mo species. **D** The EDS spectra of the trimetallic Pt-Bi/Mo/ZSM-5 catalyst. Scale bar, 50 nm (**E**) The line scan of Pt and Bi element along a particle of a bimetallic Pt-Bi/ZSM-5 catalyst. Scale bar, 25 nm. **F** The comparison of XPS spectra of Bi/ZSM-5 and Pt-Bi/ZSM-5 catalysts on Bi 4f. **G** The comparison of XPS spectra of Pt/ZSM-5 and Pt-Bi/ZSM-5 catalysts on Pt 4f.

distribution of Pt-Bi alloy, Mo ion, zeolite surface, and pores. As can be seen, the two-thirds of the ZSM-5 surface was covered by the white and shiny nanoparticles which are the infiltrated metal particles. Energy-dispersive X-ray spectroscopy (EDS) result (Fig. 2D) shows uiform elemental distribution of Mo, Pt, and Bi, where the size of Pt-Bi particle is identified to be about 30 nm, which is larger than pore size of ten-membered-ring zeolite, such as MFI (ZSM-5) with pore sizes of about 5.5 Å. Therefore, the Pt-Bi particles are located on the external ZSM-5 surface. In addition, it was demonstrated that the Mo-oxo complexes are inside the zeolite pores in 4 wt% Mo/ZSM-5 catalyst as examined by the integrated differential phase-contrast scanning transmission electron microscopy (iDPC-STEM)[23]. The line distribution of Pt and Bi shows the structure of Pt-Bi as an intermetallic alloy (Fig. 2E).

To further examine the alloy structure of Pt-Bi particles on ZSM-5, the X-ray photoelectron spectroscopy (XPS) analysis was used to study the oxidation states. We show that the shifts of binding energy for Bi 4f and Pt 4f in the single Pt, Bi, and Pt-Bi alloy in Fig. 2F, G and combine with Figure S1 to clearly inspect that the binding energy for Bi 4f in the alloy has shifted to a lower value compared to that in Bi−ZSM-5. In contrast, the binding energy for Pt 4f in the alloy has shifted to a higher value relative to Pt−ZSM-5. These changes indicate that some electrons are transferred from Pt to Bi atoms in the alloy structure and therefore there is a strong electronic interaction between the Pt and Bi. The electronic perturbation of Pt by Bi is called the ligand effect[24,25]. This effect makes Pt more "atomic like" for binding surface alkyl species more weakly to effectively form methyl radicals. This reaction is preferable for $CH_2−CH_2$ coupling instead of uncontrollable scission of the C−H bonds. It is known that ethylene is an important intermediate for the aromatics formation to enhance the benzene yield. XRD results show a weak peak appeared around $2\theta = 39.9°$ which corresponds to Pt-Bi alloy particle in both ZSM-5 supported Pt-Bi samples, which is consistent with the XPS results (Supplementary Fig. S2).

## Catalytic performance of Pt-Bi/Mo/ZSM-5 series catalysts for hydrogen and benzene co-production

To understand the synergistic effect of Pt-Bi alloy and Mo sites on enhancing catalytic activity, we compared the Pt-Bi/Mo catalysts with different Bi ratios to achieve the optimal composition and identify the role of Pt-Bi. As shown in Fig. 3A, the methane conversions, and detailed product selectivity are measured with feed gas of 95% $CH_4$/5% $N_2$ with a flow rate of 20 sccm. The definitions for conversion, product

absolute/relative selectivity, and yield are listed in the supplementary information. For the Pt/1%Mo-ZSM5 catalyst, the methane conversion and benzene selectivity had a slight improvement compared to 1%Mo-ZSM5 owing to suppression of the rate of coke formation (Fig. 3C). The increasing Bi loading from 0.3% to 0.8% could further improve the $CH_4$ conversion and benzene selectivity due to gradual formation of Pt-Bi alloy. In addition, the higher Bi loading from 1.0 to 1.5% didn't further increase activity although some improvement in catalysis stability was observed when the changes in benzene relative selectivity over the contact time were compared (Supplementary Figs. S3–8). This is because the metallic Bi covers the Pt-Bi alloy to reduce the reaction activity[26]. Thus, the optimal rate of Bi in Pt-Bi/Mo catalyst is at 0.8%.

To further reveal the role of the Pt-Bi alloy, we also compared the benchmark 2%Mo/ZSM-5 catalyst with the trimetallic 1%Pt0.8%Bi-1%Mo/ZSM-5. Figure 3B showed that the Pt-Bi/Mo/ZSM-5 had the conversion of 18.7%, which is about 25% higher than 14.9% of typical single-metal 2%Mo/ZSM. This result is mainly attributed to that the unsaturated Pt-Bi sites and the distinct active centers generated by ensemble effect promoted the formation of $CH_3^*$ on $Mo_2C$[24,27,28]. When comparing with 1%Mo, the 2%Mo/ZSM-5 has increased the conversion by 39% and the trimetallic catalyst showed 18.7% conversion, which is an even larger increase by 74%.

Figure 3C shows the respective selectivity of benzene, toluene, and $C_2$ for the catalysts with different metal compositions. As can be observed, the selectivity towards benzene was in this order: 52% (1% Mo), 61% (2%Mo), and 70% (trimetallic). The selectivity increases are in good agreement with the results on product relative selectivity (Supplementary Fig. S9). It should be noteworthy that the improvement of the selectivity is ascribed to the synergistic effect between Pt-Bi and Mo. The formed $CH_3^*$ intermediates generated from $Mo_2C$ on the external surface of zeolite can efficiently move on for C-C coupling on the surface of Pt-Bi particle for producing $C_2$ compounds which are the key intermediates for the subsequent oligomerization reaction. This synergy effect is more effective than increasing Mo content which may trigger faster catalyst degradation.

The catalyst stability was then examined by continuous operation for 8 h (Fig. 3D). Following by an initial transient activation period (~90 min), the trimetallic Pt-Bi/Mo-ZSM5 catalyst was then stabilized. The conversion decreased by 12.2% and remained at ~16%. The yield to $C_6$ achieved >10% after 8 h and the overall deactivation rate was only 17.6% while the co-product $C_2$ and $C_7$ yield was decreased by 31.0% and

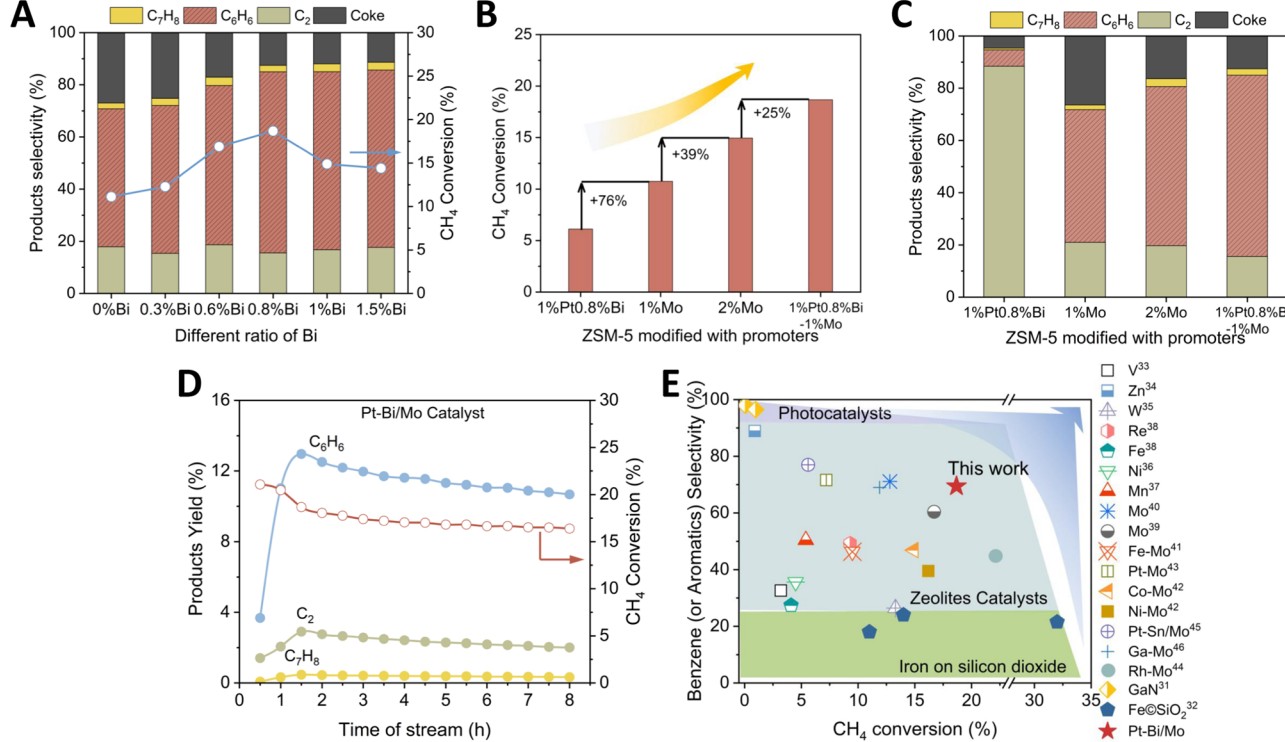

**Fig. 3 | Thermal catalytic performances of the trimetallic Pt-Bi/Mo/ZSM-5 catalysts. A** The catalytic performance of different ratios of Bismuth in Pt-Bi/Mo catalyst. **B** Measured $CH_4$ conversion and (**C**) products selectivity for different catalyst compositions tested at 710 °C, 0.95 atm $CH_4$ feed gas, and flow rate 1272 mL·$g_{cat}^{-1}$·$h^{-1}$). All data reported for catalytic performance comparison were taken at 1.5 h time-on-stream (TOS). **D** Stability of the trimetallic 1%Pt-0.8%Bi/1%Mo/ ZSM-5 catalyst over 8 h operation for observing the changes in conversion and yield. **E** A systematic performance comparison of non-oxidative MDA reactions enabled by photocatalysts and thermal chemical catalysts on aromatics selectivity and conversion.

29.3%, respectively. The Pt-Bi/Mo catalyst was then examined for nearly 50 h, and the result turned out that the catalyst conversion dropped to 13%. For comparison, 2%Mo degraded to the same methane conversion after only 7 h operation (Supplementary Fig. S10). Therefore, this composition has demonstrated acceptable durability. To examine the coke formation after the test, the spent catalysts were analyzed by TGA experiments. The result showed that each sample exhibits an initial weight loss at temperature below 250 °C attributing to the loss of humidity (Supplementary Fig. S11a)[29]. The weight loss observed between 480 °C and 680 °C corresponds to the burn-off of coke that has deposited during reaction[30]. Comparing the amount of coke formation in the Mo catalysts and the Pt-Bi/Mo catalyst (Supplementary Fig. S11b), it is evident that there is more coke formation in the former ones, which is consistent with the catalyst performance. Therefore, these results demonstrated that the addition of Pt-Bi alloy can effectively mitigate coke formation. This is probably attributed to the enhancement in the hydrogenation activity of the catalyst due to the presence of Pt-Bi and further suppressed continuous scission of C–H bonds.

The catalytic performances obtained from the catalysts developed in this work were compared with the prior results from the photocatalysts and thermochemical catalysts based on zeolite or other substrates, as shown in Fig. 3E and Supplementary Table 1. An ideal catalyst delivers high conversion and selectivity at the same time, i.e., locating at the up-right corner. Photocatalysts have superior selectivity to aromatics and can operate at low temperatures (room temperature to 350 °C). For example, the Si-doped GaN nanowires exhibited high aromatics selectivity of 96.5% at 278 K and good stability but the methane conversion is as low as 0.98%[31]. In contrast, thermochemical catalysts show much higher methane conversion but relatively lower selectivity to aromatics (e.g., the conversion: 1–32%; aromatic selectivity: 18–80%). The Fe/SiO₂ catalyst exhibited the highest methane

conversion ~32% but relatively low selectivity to aromatics (27%)[32]. The reaction temperature is about three hundred degrees higher than typical operation temperatures (e.g., 700 °C). As the mainstream catalysts for MDA reaction, the zeolite-based catalysts with embedded metals such as W, Re, Fe, and Ni are preferred options for MDA reaction[33–38], and the most efficient and commonly studied catalysts are Mo-modified ZSM-5 (Mo/ZSM-5). Methane conversions of Mo/ ZSM-5 catalysts are typically in the range of 10–13%[13,39]. By adding a second metal to form alloy such as Mo-X (X = second metal), the selectivity towards benzene can be increased by at least 60% (Fig. 3E). The stability can be also improved. Among these metal promoters, they are typically (a) transition metals (Fe, Co, Ni) that can modify the nature of carbon deposits[40,41]; (b) noble metals (Pt, Rh, Cu) that can hydrogenate to remove carbon deposits[42–44]; and (c) metals that affect the acidity of Mo/ZSM-5 catalysts (Ga, Cr)[45]. The Pt-Bi/Mo catalyst in this work clearly demonstrates one of the best performances with methane conversion ~18.7% and benzene selectivity ~69.4% at 710 °C after 90 min TOS. This catalyst has also achieved good stability, e.g., $CH_4$ conversion remained at 16% after 8 h TOS, which strongly demonstrated the potential of this trimetallic catalyst operating against coke problem.

## Effects of H₂ regeneration and operating conditions

This trimetallic catalyst has demonstrated promising resistance against the coke issue. To further investigate the potential of increasing catalyst lifetime, the catalyst regeneration experiment has been carried out. As can be seen in Fig. 4, the catalyst was first operated in MDA conditions for 6 h and then regenerated by sweeping hydrogen flow for 30 min. The conversion immediately recovered from 17.3% to 20.9%. This is a 98.6% recovery from the initial activity. Similarly, the benzene absolute selectivity was recovered from 60.2% to 69.2% after regeneration. Therefore, the conversion/selectivity after the

regeneration can almost recover or even surpass the initial activities. In addition, the regenerated activity gets better as the increase of the regeneration cycle. The previous study showed that H₂ activity promotes the reducibility of Mo oxides and formation of more proximate acidic proton-Mo sites[17], leading to the preferably internal coke formation at acidic proton-Mo sites. The hydrogen treatment can remove the internal coke easily to recover the catalytic activity via hydrocracking[46].

The effects of operating conditions on the catalytic performance were also investigated by measuring the MDA activity as function of temperature and methane concentration in feed gas. Figure S12a showed that the methane conversion and benzene yield were

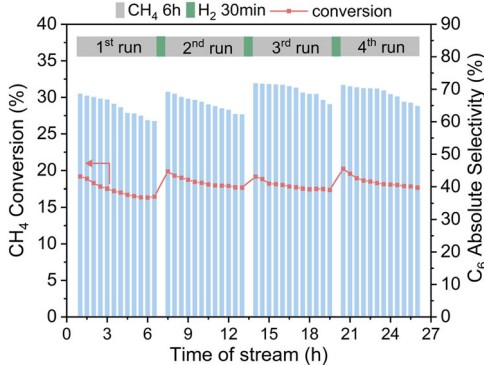

**Fig. 4 | Regeneration of the trimetallic catalyst under hydrogen environment.** The treatment under reducing condition was used to remove the coke in the catalysts efficiently for recovering the catalytic activity.

increased with temperatures up to 750 °C as measured. Furthermore, the increased methane concentration is favorable for the higher conversion and yield (Supplementary Fig. S12b). Therefore, the trimetallic catalyst should be operated at a higher methane-concentrated environment.

## Mechanism study for stability and high activity of Pt-Bi/Mo/ZSM-5 catalysts

Previous studies have demonstrated that Mo-species on the external surface predominantly form coke[39]. The Mo-carbide nanoparticles with a C/Mo ratio greater than 1.5 are more stable on an external Si site of ZSM5 than they are on an Al site, which provides the driving force for the migration of Mo-carbide from internal cages onto the external surface of the zeolite[47], as highlighted by green dashed box in Fig. 5A. Therefore, the migration of active-Mo species onto the zeolite external surface is a key factor in determining the catalyst lifetime. In comparison to the conversion deactivation rate of Mo-based catalyst, the deactivation rate of Pt-Bi/Mo catalyst decreased by 15.7% after 8 h reaction (Supplementary Fig. S13). We assume that this deactivation rate trend is related to the influence on the migration of active-Mo onto the ZSM-5 external surface during the reaction at the presence of the Pt-Bi alloy. Also, it is known that the amount of acidic proton-Mo sites and BASs are decreased as the increased ratio of Si/Al, which leads to the decreased benzene relative selectivity[48]. This influence on the mobility of the Mo species further amplifies this change of benzene relative selectivity.

To validate our hypothesis, a schematic diagram of the experiment about the different ratios of Si/Al is illustrated by Fig. 5A–C. Take the Mo/ZSM-5 catalyst as a comparison group, Fig. 5A, B showed some acidic proton-Mo sites and BASs were disappeared as the increased

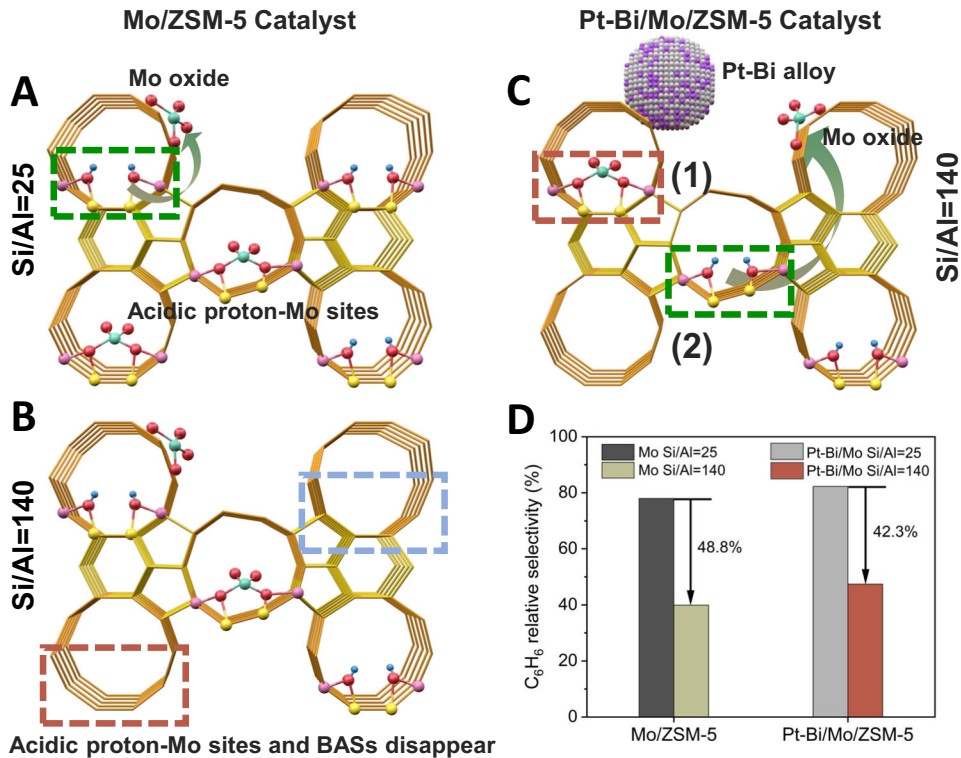

**Fig. 5 | Validation of a hypothesis for the influence on the mobility of the Mo species enabled by adding Pt-Bi alloy into the Mo/ZSM5 catalyst. A, B** Schematic diagrams of the changes in acidic proton-Mo sites and Brønsted acid sites (BASs) as the ratio of Si/Al increases from 25 to 140 on Mo/ZSM-5 catalyst. The red and blue dashed boxes highlighted that acidic proton-Mo sites and BASs disappeared, respectively. Mo species move from BASs to the external surface of ZSM-5 as green dashed box is highlighted. **C** Schematic diagram of two possible cases for Pt-Bi/Mo/ZSM-5 catalyst: (1) Pt-Bi alloy reduces the mobility of Mo oxides (red dashed box) or (2) Pt-Bi alloy promotes Mo oxides migration to the zeolite surface (green dashed box). **D** Benzene relative selectivity for Mo and Pt-Bi/Mo catalysts with different ratios of Si/Al. Color scheme: Mo (cyan-blue), Bi (purple), Si (yellow), Al (pink), H (blue), O (red) and Pt (white).

ratio of Si/Al from 25 to 140 responsible for the decreased benzene relative selectivity. For Pt-Bi/Mo/ZSM-5 catalyst, it is hypothesized that there are two possible situations (Fig. 5C). First, the Pt-Bi alloy might enhance the retention of the Mo species inside the zeolite channels during reaction, as highlighted by the red dashed box, leading to the alleviated the attenuation of $C_6H_6$ selectivity as the increased ratio of Si/Al. Second, the Pt-Bi alloy may promote the migration of acidic proton-Mo sites, described by the green dashed box, which promotes the attenuation of $C_6H_6$ selectivity. It was found that the attenuation rate of $C_6H_6$ relative selectivity (42.3%) for Pt-Bi/Mo/ZSM-5 was lower than that for Mo/ZSM-5 catalyst (48.8%), as illustrated in Fig. 5D, which indicates that Pt-Bi alloy alleviates the mobility of some active Mo species from cages to the outer surface. Furthermore, it is known that the low-angle X-ray intensities are sensitive to the presence of any species inside the ZSM-5 channels[49,50]. The low-angle X-ray diffraction (XRD) pattern showed that the peak intensity between 7.9° and 9° of Pt-Bi is close to the peak intensity of ZSM-5 (Supplementary Fig. S14). The obvious decrease of peak intensity for Mo and Pt-Bi/Mo catalysts implies the migration of Molybdenum oxide into the channels. Compared with Mo, the peak intensity between 7.9° and 9° of Pt-Bi/Mo is lower, which might indicate the mobility of some Mo species in the pores of ZSM-5 is alleviated. This result is consistent with the Brunauer-Emmett-Teller (BET) results and STEM results in Supplementary Fig. S15. As can be observed in Supplementary Table 2, the microporous texture properties ($S_{micro}$ and $V_{micro}$) of Pt-Bi/Mo catalyst decreased at the addition of the Pt-Bi particles. This is likely due to strong interaction between Pt-Bi particles and external BASs on the catalyst blocking some of its pore structures[51], further facilitating the retention of Mo species in the zeolite cage.

There are two major factors that determine the reaction kinetics: the number of active sites for catalysis and activation energy ($E_a$). The temperature programmed desorption (TPD) is an effective approach to analyze the kinetics of desorption for different molecular species. The area under the TPD profile is proportional to the amount of adsorbate originally adsorbed, in other words, to the surface coverage[52]. As can be seen in Fig. 6A, there was only one $CH_4$ desorption peak observed for 1%Mo/ZSM-5 and 2%Mo/ZSM-5 at 178 °C. In contrast, there were two additional distinct and broader desorption peaks for 1%Pt0.8%Bi-1%Mo/ZSM-5 at higher temperatures spanning approximately from 250 to 400 °C. This result clearly suggests that the desorption peaks at 275 °C and 347 °C were attributed to the addition of Pt-Bi alloys onto Mo-ZSM-5 substrate. By further analysis on the relative peak area in the order of Pt-Bi/Mo > 2%Mo > 1%Mo, the result indicates that $CH_4$ surface coverage of Pt-Bi/Mo was increased by 55% for 2%Mo/ZSM-5 and by 56% for 1%Mo/ZSM-5, which is consistent with methane conversion results as discussed before.

Temperature programmed reduction (TPR) was further used to confirm the reduction behaviors of the Mo phase and the influence of Pt-Bi alloy on the reducibility of the Mo/ZSM-5 catalyst. This is closely related with the ethylene formation as an intermediate for the reaction[53,54]. Figure 5B presented that the reduction of the Mo catalyst resulted in two main peaks at 175 °C (circle) and 534 °C (square), which were from the reduction of $MoO_3$ to $MoO$, and the reduction of $MoO$ to metallic Mo, respectively. Upon addition of Pt onto the Mo/ZSM-5, there were changes in two peak positions (shifting from 175 to 123 °C, 534 to 523 °C) while the first reduction peak shifted to lower temperatures obviously. This result indicates the addition of the Pt promoted the reducibility of the molybdenum oxides to provide activated hydrocarbon atoms. Because Mo oxide nanostructures reduced to carbide species ($Mo_xC_y$) or oxycarbide ($MoO_xC_y$) when $CH_4$ was the only reactant. As the $CH_4$ molecule approached the Mo center of the $Mo_xC_y$ cluster, the electrons of the C−H bond were partly transferred

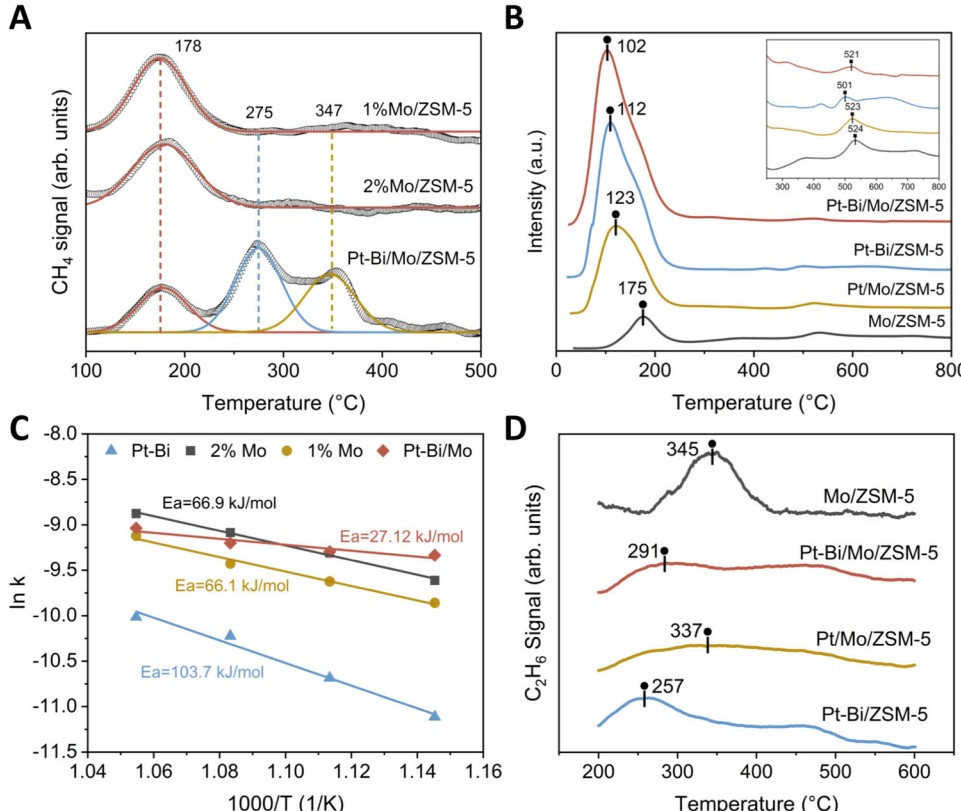

**Fig. 6 | Fundamental characterizations on studying the mechanisms for the facilitated reaction kinetics. A** Surface desorption investigation by $CH_4$-TPD in 99% $CH_4$. **B** TPR study of the prepared catalysts under 5% $H_2$/Ar to examine the formation of $MoC_x$. **C** Activation energies for different catalysts are determined by Arrhenius relationship. **D** The $C_2H_6$-TPD pattern in 99% $C_2H_6$ for studying the $C_2$ desorption behaviors of the catalysts.

to the vacant d-orbitals of the Mo atom just above the Fermi level, which subsequently caused the C−H bond activation and the formation of activated hydrocarbon atom[55]. Additionally, the lower temperature makes the reduction easier[56], therefore the presence of the Pt promotes the generation of $Mo_xC_y$ for subsequent reactions. Interestingly, the introduction of Bi to Pt/Mo showed a further decrease in the reduction temperature of Molybdenum oxide species. The first peak is associated with the reduction of $MoO_3$ to $MoO$, shifting from 123 to 102 °C, which can be explained by the promotion of Pt-Bi bimetallic on the reducibility of Mo due to the ligand effect. The d-band center of Pt-based bimetallic alloys is facilely tuned to change electronic properties of bimetallic alloys for improving the adsorption and activation of reactants[27].

To directly describe the reaction kinetics, the activation energies were calculated for comparison to study the synergistic effect. The Napierian logarithm of reaction rate constant (Arrhenius curve) as a function of reaction temperature reciprocal was plotted in Fig. 6C. From the fitting report in Supplementary Table 3, the residual sum of squares and the Pearson's r for all reactions are all smaller than 0.02 and −0.95 respectively, which indicates that the linear fitting is reliable[57]. As we can see, the activation energy for 1%Mo/ZSM5 (66.1 kJ/mol) was very close to that of 2%Mo/ZSM5 (66.9 kJ/mol). Therefore, the different active loadings in the catalysts have not impacted the barrier energy of the reaction because the same composition has the same transition state for breaking C-H bonds in methane. In a clear contrast, the Pt-Bi catalyst has a larger activation energy of 103.7 kJ/mol, yet the Pt-Bi/Mo catalyst has a much smaller activation energy of 27.12 kJ/mol. This result strongly demonstrates the synergy between Pt-Bi and Mo for enhancing the reaction kinetics and verifies our purposed mechanism that the initial $CH_4$ activation mainly happens on the Mo species, and the subsequent C-H activation and $C_2H_4$ formation achieve on Pt-Bi particles. In addition, it is in good agreement with $CH_4$ conversion results in Fig. 3B.

To investigate the mechanism responsible for the increased benzene selectivity, the $C_2H_6$-TPD experiments were carried out to analyze the desorption behavior of the $C_2$ intermediates during the reaction. It should be noted that the higher desorption temperature reflects a stronger bonding between the adsorbate species and surface[52], which can potentially lead to deep dehydrogenation of the $C_2$ intermediates. Figure 6D shows that the desorption temperature of Mo/ZSM-5 is the highest, indicating the further dehydrogenation of

$C_2H_6$ (forming $CH_2$, $CH$ and $C$ species) on $Mo_2C$ sites. In comparison, the desorption temperature (291 °C) of Pt-Bi/Mo/ZSM-5 was lower than 345 °C for Mo/ZSM-5, which should be ascribed to Pt-Bi particle facilitating the desorption of surface alkyl species on Mo-C site of the zeolite surface to form methyl radicals. These $CH_3^*$ are further transferred to the surface of Pt-Bi particles for preferential C-C coupling and subsequent oligomerization of $C_2$ intermediates on BASs instead of continuous cleavage of C-H bonds for improving the selectivity of $C_6$ products.

## DFT simulations for understanding synergy between Pt-Bi alloy and Mo species

Experimental results and characterizations suggest that when the Pt-Bi alloy is introduced in Mo/ZSM-5, both the catalyst's reactivity and durability improve, indicating that Pt-Bi, as a promoter, assists the transformation of $CH_4$ into the $C_2$ intermediate by alleviating coke formation so that the catalyst lifetime is prolonged and catalytic activity is enhanced. This hypothesis is then investigated using DFT so that the roles of Pt-Bi alloy and the Mo species can be understood.

All surface intermediates except $CH_4$, that is, $CH_3$, $CH_2$, $C_2H_4$, and H bind more strongly on the $Mo_2C$ (001) than on Pt-Bi (111) (Supplementary Table 4), thus, making the initial $CH_4$ activation and conversion energetically favorable. These results agree with past studies in literature, where the Mo/ZSM-5 is a known effective catalyst for the MDA reaction. Moreover, our experiments also suggested that Pt-Bi does not display notable reactivity toward methane conversions. A closer examination indicates that the Mo species is responsible for the strong interactions between H and carbon ($CH_x$ and $C_2H_4$) species. In the Pt-Bi alloy, H and carbon species prefer to bind with the Pt species. Also, $CH_3$ and $CH_2$ favor the site that satisfies the electronic valency of the central carbon atom, i.e., $CH_3$ prefers the Pt top site, while $CH_2$ is likely located between two Pt atoms.

As shown in Fig. 7, the activation of the first $CH_4$ C−H bond is highly endothermic on all surfaces due to entropy loss, with an energy barrier of 0.5 ~ 0.6 eV. Because of the strong binding of $CH_3$ and H on $Mo_2C$ (001), the reaction is much less endergonic than on Pt-Bi (111) (by ~0.9 eV). The subsequent decomposition of $CH_3$ is distinct between the two surfaces, where $CH_3$-to-$CH_2$ becomes exothermic (-0.54 eV) on $Mo_2C$ (001) but remains slightly endothermic (0.14 eV) on Pt-Bi (111). Afterward, the formations of gas-phase $H_2$ and adsorbed $C_2H_4$ via recombinations of H and $CH_2$ on $Mo_2C$ (111) become endothermic.

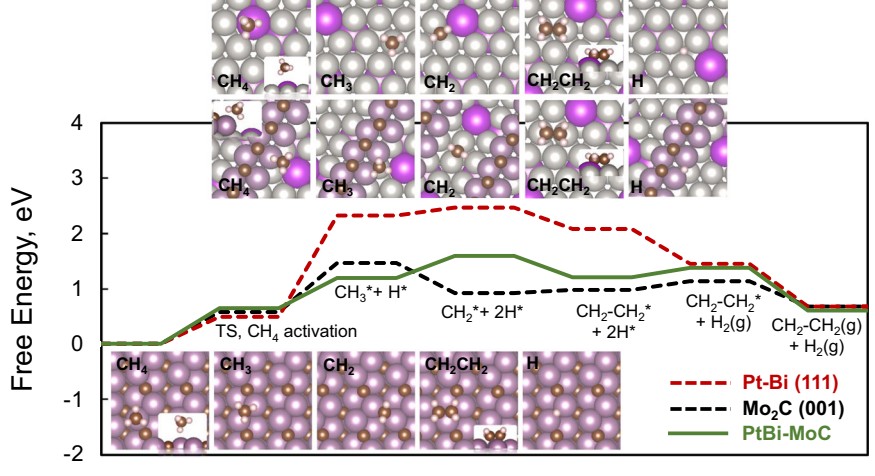

**Fig. 7 | Energy landscape and DFT optimized configurations where conversion of $CH_4$ into $C_2H_4$ on Pt-Bi (111) (red dashed lines), $Mo_2C$ (001) (black dashed lines), and Pt-Bi-MoC (solid green lines) surfaces.** The free energy changes for $CH_4$ dissociation, $H_2$ and $C_2H_4$ desorption are estimated at 710 °C and 1 bar. The configurations of adsorbed $CH_4$, $CH_3$, $CH_2$, $C_2H_4$, and H are displayed as inset figures. Color scheme: Mo (purple), C (brown), Bi (magenta), and Pt (gray).

In contrast, these steps are much more thermodynamically favorable on Pt-Bi (111), as depicted by the downward shifting free energy landscape.

Although the final $C_2H_4$ desorption step is exergonic on both surfaces, the free energy surfaces describe very different pathways on separated $Mo_2C$ and Pt-Bi catalysts. The initial activation and conversion of $CH_4$ on $Mo_2C$ (001) benefits from the strong bindings with the Mo sites with respect to the prohibitive thermodynamic barrier on Pt-Bi. Nevertheless, once $CH_4$ is activated, both $H_2$ and $C_2H_4$ formations are competitive thanks to the moderate binding with the Pt site. When $Mo_2C$ and Pt-Bi alloy are combined, the $CH_4$-to-$C_2H_4$ process can take advantage of such distinct catalyst behaviors. Based on the Pt-Bi-MoC composite model, the initial $CH_4$ activation pathway was determined using the CI-NEB method. We deliberately chose the supported $Mo_2C$ front edge owing to the demonstrated reactivity for $CH_4$ conversions. Such edge site shall mimic the interface established by the two primary catalytic materials due to its proximity to the Pt-Bi underneath. An energy barrier of ~0.55 eV was obtained (Supplementary Fig. S16), which is in line with that on $Mo_2C$ (001). As discussed previously, $CH_4$ activation is mainly hindered by the thermodynamics of this elementary step. We noted that the reaction free energy is reduced by ~0.27 eV, lower than both Pt-Bi (111) and $Mo_2C$ (001). Such a reduction benefits from the edge Mo sites that anchor and stabilize the $CH_3$ group upon the C–H bond activation, as evidenced by the potential energy valley. We also observed that, for $CH_3$ to migrate across the interfacial boundary onto Pt-Bi, a low energy barrier of ~0.4 eV is needed. Under the MDA temperature, such migration would be facile. On the Pt-Bi support, the conversion of $CH_3$ into $CH_2$ is endothermic (0.40 eV). The subsequent $C_2H_4$ formation will follow a similar path to that on Pt-Bi (111). With the composite model, the overall free energy landscape has acquired the behaviors from both its components. The initial $CH_4$ activation resembles that of $Mo_2C$ (111), and the subsequent C-H activation and $C_2H_4$ formation happens on Pt-Bi (111). Hence, Pt-Bi-MoC not only retains, but also somewhat improves the reactivity. As the reaction proceeds, carbonaceous species migrating onto Pt-Bi will more likely continue to form $C_2H_4$, which eventually desorbs from the system easily instead of occupying and blocking the active sites.

## Discussion

Designing highly active and durable thermochemical catalysts for MDA reaction to produce hydrogen and liquid aromatics is very critical for upgrading natural gases to produce higher-value added chemicals. This work successfully demonstrated the synergetic interaction between bimetallic Pt-Bi alloy cluster and Mo to significantly promote catalytic performance and lifetime of catalyst. The catalysts were systematically synthesized and tested for evaluating the fundamental mechanism of the interactive effect of these two essential components for understanding each underlying effect. On the other hand, the Pt-Bi/Mo/ZSM-5 catalyst with the bifunctional mechanism first activates the C–H bond of methane on $Mo_2C$ of the external ZSM-5 to form $CH_3^*$, further C-C coupling to produce $C_2$ intermates on the surface of Pt-Bi alloy, and then aromatizes them to benzene and other aromatic compounds on the BASs. Meanwhile, active-Mo sites in the pores of ZSM-5 also perform dehydrogenation and oligomerization of methane. The 1%Pt-0.8%Bi/1%Mo/ZSM5 catalyst showed 18.7% conversion, 69.4% benzene absolute selectivity at 710 °C for 95% $CH_4$ flow stream. The catalyst also showed good stability with 17.6% yield reduction and 12.2% conversion decrease respectively, which are better than conventional Mo-based catalysts. A series of fundamental characterizations have indicated that the synergy between the bimetallic Pt-Bi alloy and Mo species plays the key role. First, the activation energy is significantly decreased to facilitate the reaction kinetics. Along with the conventional reaction pathway via the Mo-site in the zeolite cage, the trimetallic catalyst clusters on the zeolite surface add another pathway for controllable reaction chain, e.g., adsorption, dissociation, coupling, and aromatization. In this process, the Mo specie, Pt-Bi alloy, and BASs work collaboratively to enhance the reaction rate. This hypothesis has been partially confirmed by the TPD study. The proper desorption temperature is favorable for the fast transfer of methyl radicals to Pt-Bi surface for preferential C–C coupling instead of continuous cleavage of C–H bonds for improving the selectivity of $C_6$ products.

The DFT calculation results have revealed the energy landscape favorable for the reaction due to the synergy effect. Because of the strong binding of $CH_3$ and H on $Mo_2C$ (001), the reaction is much less endergonic than on Pt-Bi (111) (by ~0.9 eV). The subsequent decomposition of $CH_3$ is distinct between the two surfaces, where $CH_3$-to-$CH_2$ becomes exothermic (−0.54 eV) on $Mo_2C$ (001) but remains slightly endothermic (0.14 eV) on Pt-Bi (111). Afterward, the formations of gas-phase $H_2$ and adsorbed $C_2H_4$ via re-combinations of H and $CH_2$ on $Mo_2C$ (111) become endothermic. In contrast, these steps are much more thermodynamically favorable on Pt-Bi (111), as depicted by the downward shifting free energy landscape.

The synergy effect of Pt-Bi is also very important to mitigate the coke issue. As confirmed by the experimental study, the amount of acidic proton-Mo sites is decreased as the increased ratio of Si/Al in zeolite from 25 to 140 responsible for the decreased benzene relative selectivity. The Pt-Bi alloy might enhance the retention of the Mo species inside the zeolite channels during reaction so that to suppress the attenuation rate of $C_6H_6$ selectivity. The $H_2$ regeneration process was used to address the Pt-Bi/Mo catalyst deactivation. The methane conversion was instantly recovered from 17.3% after 6 h operation to 20.9% after regeneration, and this is a 98.6% recovery of the initial activity from the sample after the durability test.

In summary, this work offers a trimetallic catalyst for MDA reaction and its synergy effect between Pt-Bi and Mo gives some insights into the design of highly active and durable thermochemical catalysts.

## Methods
### Preparation of MDA catalysts
The catalysts were synthesized by the incipient wetness impregnation approach. First, the Mo/H-ZSM-5 baseline catalyst was prepared by infiltrating Mo salt solution into H-ZSM-5 zeolite (CBV 5524G, Si/Al = 25, Zeolyst International) to obtain 1 wt.% solid loading. Ammonium heptamolybdate $[(NH_4)_6Mo_7O_{24}\cdot4H_2O]$ was dissolved in deionized water to form a solution, followed by sonication for 5 min. The salt solution was sequentially added to H-ZSM-5 powders with 200 μl for each step by pipette. After complete mixing, the catalyst was dried at 110 °C in air for 24 h and then calcined in flowing air at 700 °C for 2 h with a ramp rate of 3 °C min$^{-1}$ to form Mo/H-ZSM-5 catalyst. Based on this composition, the tri-element catalysts were prepared by further adding platinum and bismuth onto Mo/H-ZSM-5 samples with the solutions of tetraammineplatinum nitrate $Pt(NH_3)_4(NO_3)_2$ (Alpha Aesar, 99.9%) and bismuth nitrate pentahydrate $Bi(NO_3)_3\cdot5H_2O$ (Sigma-Aldrich, 99.999%). The Pt and Bi loading were controlled at 1.0 wt.% and 0–1.5 wt.%, respectively. The powders were also dried at 110 °C and then calcined at 700 °C for 2 h to form the final catalysts. Another set of 2.0 wt.% Mo/ZSM-5 and Pt (1.0 wt.%)-Bi (0.8 wt.%)/H-ZSM-5 catalysts were also prepared by the same procedures for comparison.

### Characterizations of synthesized catalysts
The crystal structure and element valence states were examined by X-ray diffraction (XRD, M-D8 Advances) using Cu-filter Cu Kα radiation (45 kV, 200 mA), and diffraction patterns were collected within the 2-theta range from 5 to 60° with a scanning rate of 2° min$^{-1}$. The physisorption properties of catalysts, including surface area, pore size, and pore volume were calculated by the BET equation. Prior to the measurement, all samples were degassed under vacuum for 12 h at 350 °C to remove surface humidity. XPS was utilized to observe the oxidation states of Pt and Bi in alloys to determine the structure and morphology. The measurements were performed using a PHI VersaProb+ III

(Physical Electronics) equipped with a monochromatic Al Kα source operated at 15 keV and 50 W and a hemispherical energy analyzer. The analyzer-to-source angle was 50°, whereas the emission angle was 45°. A pass energy of 50 and 25 eV was set for the survey and narrow scans, respectively, and the C 1s peak of adventitious carbon was set at 284.8 eV to compensate for any charge-induced shift. TGA profiles were recorded on a NETZSCH STA 449 F3 *Jupiter®*. The flow rate of air was 30 ml/min. The coked catalyst was heated from 302 to 1223 K in an air stream at a heating rate of 10 K/min.

### Mechanism studies
To understand the improvements in catalytic activity, temperature programmed reduction (TPR) was used to investigate the reduction properties of the catalysts. 100 mg of each sample in the quartz tube was first heated at 200 °C for 1 h at the heating rate of 10 °C min$^{-1}$ under an Ar gas (30 mL min$^{-1}$). After the temperature was cooled down to room temperature (<50 °C) under Ar atmosphere, the temperature went up from 30 to 750 °C in a 3% H$_2$/N$_2$ mixed gas (20 mL min$^{-1}$), rising linearly at the rate of 10 °C min$^{-1}$. The information on H$_2$ content was collected by a high precision quadrupole mass spectrometer (QMS; Inficon Transpector 2).

To explain high conversion and benzene selectivity, TPD was performed to continuously monitor the desorption of CH$_4$ and C$_2$H$_6$, and the activation energy was calculated. For the CH$_4$-TPD experiments, 100 mg of each sample was pretreated at 200 °C for 1 h in Ar (30 mL min$^{-1}$) and then cooled down to room temperature for immersing in a 99% CH$_4$/N$_2$ mixed gas (20 mL min$^{-1}$) for 2 h to fully absorb methane molecules. Then the quartz tube was purged with Ar and held for 1 h at 100 °C to remove physically accumulated CH$_4$ from the catalyst surface. Finally, the samples were heated from 100 to 750 °C in 99% CH$_4$/N$_2$ mixed gas at 5 °C min$^{-1}$ and maintained for 10 min to acquire CH$_4$-TPD profiles. Similarly, for the C$_2$H$_6$-TPD experiments, 100 mg of each sample was first heated at 200 °C for 1 h under Ar and lowered to room temperature, then exposed to C$_2$H$_6$ flow for 2 h until saturation. And then, the gas was switched to Ar and stayed for 1 h. The temperature was then ramped to 750 °C at 10 °C min$^{-1}$ and held for 10 min until complete desorption of C$_2$H$_6$. The information on CH$_4$ and C$_2$H$_6$ desorption content was collected by mass spectrometer (QMS, Inficon Transpector 2). For the calculation of activation energy, the operating conditions were: 600–675 °C and 95%CH$_4$. The collected data were taken on every 25 °C and at 1.5 h time-on-stream (TOS).

### Catalytic performance measurements
The catalytic performance of the as-synthesized catalysts was evaluated by a fixed-bed reactor approach with a 7 mm inner diameter and a length of 450 mm. The catalyst bed consisted of 500 mg of sample, held in place between two quartz wool plugs. The standard operating conditions were: 710 °C, 95%CH$_4$, 0.50 g catalyst and 1272 mL·g$_{cat}$$^{-1}$·h$^{-1}$ contact time. In typical cases, following an initial transient period, the catalyst exhibited stable performance after 1.5 h. Unless stated otherwise, all data reported in our work for catalytic performance comparison were taken at 1.5 h TOS. A blank experiment using the pure ZSM-5 without any Pt, Bi, or Mo loading was tested under standard operating conditions, with methane conversion less than 0.01%. With nitrogen as an internal standard, the products leaving the reactor were analyzed using a gas chromatograph-mass spectrometer equipped with a flame ionization detector and high-performance ion source (Shimadzu GCMS-QP2020NX). The equations (Supplementary Eqs. 1–4) were used to calculate methane conversion, product relative and absolute selectivity, and yield for the detectable products (ethane, ethylene, benzene, and toluene). The testing results were repeated more than two times to have data deviation less than 2%.

### Computational methods and models
All spin-polarized density functional theory (DFT) calculations were performed using VASP[58,59], where the wave functions are formulated according to the Bloch theorem. The projector augmented wave method was used to approximate the wavefunctions near the core region[60]. The electronic exchange-correlation energy term is accounted for using the GGA Perdew–Burke–Ernzerhof functional[61].

Based on the assumption that the initial conversion of CH$_4$ into the precursor of benzene (i.e., C$_2$H$_4$) takes place at the metallic sites, the primary active site configurations for the CH$_4$-to-C$_2$H$_4$ conversion were constructed using Mo$_2$C and Pt-Bi bulk crystals. The structure of Mo$_2$C (mp-1552), with the lowest formation energy, was extracted from the Materials Project. The Pt-Bi alloy was constructed using a (2 × 2 × 2) face-centered cubic Pt supercell (Supplementary Fig. S17). Ionic relaxations of these bulk structures were carried out using a cut-off energy of 520 eV and an automatically generated 6 × 6 × 6 k-point mesh based on the Monkhorst-Pack scheme[62]. The convergence criterion for the self-consistent energy loop is set to be $1 \times 10^{-6}$ eV, and the ionic steps stop when the forces on each relaxed atom are less than 0.02 eV/Å. The orbital partial occupancy was determined using the Methfessel-Paxton method[63]. The lattice constants are relaxed to $a = 4.74$ Å, $b = 5.23$ Å, and $c = 6.06$ Å for the orthorhombic Mo$_2$C, which compare well with literature values[64]. The lattice constants for Pt-Bi become a = 8.12 Å and b = c = 8.14 Å. Due to the large Bi atomic radius, the cell expands by ~2.3% compared to pure Pt bulk ($a = b = c = 7.95$ Å). Lattice strain has a pronounced effect on metal surfaces. Nørksov and coworkers[65,66] showed that the binding strengths of surface adsorbates increase and molecular activations (e.g., CH$_4$, CO, H$_2$) become easier as the d-band center shifts to a higher energy level when the lattice expands. Hence, the effect of Pt-Bi alloy on the CH$_4$-to-C$_2$H$_4$ conversion will be understood by examining its elementary steps.

Surface reaction steps depicting a simplified CH$_4$-to-C$_2$H$_4$ conversion process are modeled using DFT calculations on Mo$_2$C semi-periodic Pt$_3$Bi employing Mo$_2$C (001) and Pt-Bi (111) slabs. A cutoff energy of 400 eV and a 4 × 4 × 1 k-point mesh were used, while other parameters remained unchanged. Moreover, a composite model derived from Mo$_2$C and Pt-Bi was employed to investigate a possible synergistic pathway facilitating the CH$_4$-to-C$_2$H$_4$ conversion in the co-catalyst setting. A one-dimensional (1D) stoichiometric Mo$_2$C wire was placed on top of the Pt-Bi (111) surface (denoted as Pt-Bi-MoC in Supplementary Fig. S17). This configuration was chosen because of the similarity of lattice parameters between Mo$_2$C and Pt-Bi. The *front* side of the wire was used to study the surface chemistry.

The transition state search employed the standard climbing-image nudged elastic band (CI-NEB) method[67]. The final transition states were determined by refining the structure using the dimer method[68] and vibrational frequency analyses.

## Data availability
All data supporting the findings of this study are available within the article, as well as the Supplementary Information file. All other data supporting the findings of the study are available from the corresponding authors if asked for.

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

## Acknowledgements

This work is supported by the Laboratory Directed Research Development Program in Idaho National Laboratory under contract no. DE-AC07-05ID14517. C.D. and H.D. would like to thank the USDOE DE-FE0032235 for the funding support. H.D. would like to thank the startup research grant from the University of Oklahoma. The DFT computational support was provided by the Beocat Research Cluster at Kansas State University is funded partly by NSF grants CHE–1726332; and the High-Performance Computing Center at Idaho National Laboratory with funds by the Office of Nuclear Energy of the U.S. Department of Energy and the Nuclear Science User Facilities under Contract No. DE-AC07-05ID14517.

## Author contributions

D.D., P.D and Hanping.D. conceived, designed, and supervised the project. P.Z. conducted catalyst test, data analysis and prepared the manuscript. W.B. synthesized the catalysts. B.L. and Hao.D. performed the DFT calculation. X.H. performed the BET characterizations. S.S. and F.L. performed the XPS characterizations. Hanping.D. and C.D. revised the manuscript. L.W. supervised experimental and computational efforts. All authors commented on the final version of the manuscript.

## Competing interests

The authors declare no competing interests.
