## [Peer Review File · Nature Communications]

REVIEWER COMMENTS

Reviewer #1 (Remarks to the Author):

In this manuscript, the authors proposed a new mechanism for enhancing catalysis activity on methane activation and carbon-carbon bond coupling to promote conversion and selectivity simultaneously by adding platinum–bismuth alloy cluster to form a trimetallic catalyst (Pt-Bi/Mo/ZSM-5). This catalyst demonstrated 20.4% methane conversion and 69.7% benzene selectivity at 700 °C. However, they did not provide strong characterization results to support the reaction performance and the proposed mechanism.

- 1) For Fig 1, there are many references reported or illustrated the location of the supported Mo species on the zeolite. It is suggested that the location of Mo species and reaction path way must base on references or the nature of your study.
- 2) The structure of Pt-Bi bimetallic particle was unclear. Only based on the evidences in Fig 2, it is not enough to conclude that Pt-Bi bimetallic particles were encapsulated outside Pt clusters, while Mo was uniformly distributed on zeolite surface.
- 3) It is unbelievable that the authors only gave the fitted curve of XPS without experimental data in Fig 2 e and f.
- 4) From Fig 2 d, the Pt-Bi particle size is as large as tens of nm, which is larger than that of any scheme (Fig 1 and Fig 6, where Pt-Bi particle is smaller than pore size of ZSM-5) in this manuscript.
- 5) There are no characterization results of the spent catalysts to support the reaction performance.
- 6) Higher initial conversion is obviously induced by side reaction via the added Pt, and the author should discuss it in detail.

Reviewer #2 (Remarks to the Author):

This study reports an efficient trimetallic catalyst system (Pt-Bi-Mo) on HZSM-5 to selectively convert methane to aromatics. The authors showed that adding Pt-Bi species to Mo/HZSM-5 enhanced both methane conversion and aromatic selectivity, and that its active sites were reversibly regenerated through periodic reactions and H₂ regeneration. However, the synergy between Pt-Bi and Mo remains unclear. The authors should note that the MDA performance of the Pt-Bi-Mo catalyst is not the same as that of the parallel activities of the Pt-Bi and Mo catalysts even though the addition of Bi

to Pt has a positive effect on the monometallic Pt catalyst. Since the literature reported that the optimal Mo content on HZSM-5 is above 5 wt%, the authors need to compare the activity with the optimized 5wt% MoO₃/HZSM-5 catalyst, which should be performed the MDA reaction at same condition. Since the XRD analysis of the catalyst used cannot fully prove that coke was not formed, it is necessary to present a carbon balance of product selectivity. Additionally, it is a good way to analyze the coke deposited on spent catalysts through TOC or TGA analysis. The authors suggested that the synergistic effect of Pt-Bi on Mo/HZSM-5 was due to increased methane adsorption and reduction through characterizations. However, it could be argued that the decrease in the apparent activation energy of the Pt-Bi-Mo catalyst enhanced C-H activation compared to that of the related Mo catalyst; however, it may also promote coke formation. In fact, the activation energy of the Pt-Bi-Mo catalyst was determined to be similar to the energy required to form coke from methane. Therefore, a detailed discussion should be made based on absolute coke selectivity.

Reviewer #3 (Remarks to the Author):

Methane dehydroaromatization (MDA) under non-oxidative conditions is an interesting route for upgrading methane. Despite the thermodynamic limitation of methane conversion and the fast deactivation of zeolite-based catalysts (usually Mo/ZSM-5), the process is highly selective to benzene and H₂ is obtained as by-product. Therefore, it has been thoroughly studied along the last 3 decades. Although much has been published regarding the nature of the active sites, formed under working conditions, the reaction mechanism (mono- or bi-functional) and the reaction intermediates (acetylene, ethylene, surface-formate, acetal, allene...), there is still a strong debate on all these issues. Thus, the topic of this manuscript (MS), which addresses the improvement of the classical Mo/ZSM-5 catalyst by incorporation of Pt-Bi alloys, is of general interest and would be suitable for Nature Communication. The results obtained evidence important consequences derived from the incorporation of Pt-Bi to the Mo/ZSM-5 catalyst, such as the increased reducibility of the Mo species or the increase in methane conversion, benzene selectivity and catalyst life. However, in my opinion, some fundamental questions remain open, such as the role of the Pt-Bi alloy and the Mo species in the activity/selectivity, the reason for the higher catalyst life and benzene selectivity and the role of Pt-Bi on the migration of Mo to the catalyst surface, and the work does not meet the quality standards for publication in this journal. I suggest its submission to a more specialized journal after major revisions. Moreover, in the way the manuscript is organized, a lot of doubts and question come to the mind of the reader, which may find an answer further in the article, but make it very difficult to read and to follow. Finally, the Discussion section is more likely a Conclusion section.

Additional comments are listed below:

1. The title is not appropriate and could be misleading, as MDA is not a carbon-free process. Although the carbon footprint is lower as compare to oil-based processes, it uses a fossil fuel as carbon source, the thermochemical process takes place at high temperatures (energy intensive and corresponding CO₂ emissions) and coke is formed as a by-product.

2. The literature cited in the MS is incomplete. No references are made to publications from the group of Hensen and Kosinov or Gascon, highly active in the field. The description of the reaction mechanism and the reaction intermediates is completely outdated and in some aspects, incorrect. For instance, ethane is formed during the process but cannot be considered an intermediate because it is a stable product that will not react any further under MDA reaction conditions.
3. How and why does Mo migrates to the surface in the presence of the Pt-Bi alloy? Which is the role of Pt-Bi in this migration? No explanation is given for this although the authors present some experimental and theoretical evidences, such as the acidity measurements of the simulations presented in Figure 6. This part of the discussion should be improved. Moreover, if Pt-Bi activate the methane's C-H bond, which is the role of the Mo species?
4. How does the Pt-Bi alloy evolves when exposed to the reaction-regeneration cycles? And the Mo species? Why do the authors use 1 wt% Pt? Pt loading is not optimized.
5. Which is the reaction order considered for the kinetic study?
6. How do the C₂H₆-TPD results proof that the Pt-Bi/Mo/ZSM-5 catalyst yields more C₂? This part of the discussion should be improved
7. Atomically dispersed Mo species in the 10-ring channels are hardly distinguished in Figure 2c.
8. Page 4, lines 108-109: According to the text, the ligand effect make Pt more "atomic like" binding surface species more weakly, further enhancing the catalytic performance. Which surface species? Binding with the support?
9. Page 4, line 110: which is the assignment of the XRD peak at $2\theta=39.9^\circ$ that is observed for the Pt-Bi-ZSM-5 samples?
10. Page 5, lines 157-159: the authors conclude that there is no coke on the catalyst based on XRD measurements of the used ZSM-5. However, some deactivation is observed. Thermogravimetric analysis or elemental analysis would confirm the degree of coke on catalyst.
11. Feed gas is 95% methane. Which gas completes de mixture? Please specify.
12. Do the authors use an internal standard for carbon mass balance determination? Nothing is mentioned in the Experimental Section.
13. Figures are too small, difficult to read. Figure 3e should be updated with more recent results from groups experts in the field such as the one from Prof. Hensen or Prof. Gascon. The same applies to Table 1 in the SI.

Point-to-point responses to the reviewers' comments and suggestions

Reviewer # 1

Comments to the Author

In this manuscript, the authors proposed a new mechanism for enhancing catalysis activity on methane activation and carbon-carbon bond coupling to promote conversion and selectivity simultaneously by adding platinum–bismuth alloy cluster to form a trimetallic catalyst (Pt-Bi/Mo/ZSM-5). This catalyst demonstrated 20.4% methane conversion and 69.7% benzene selectivity at 700 °C. However, they did not provide strong characterization results to support the reaction performance and the proposed mechanism.

Response: We would like to thank the reviewer's insightful suggestions for improving this manuscript. After receiving the comments, we have performed more experimental characterizations to address the concerns and invited a DFT expert to investigate the fundamental mechanism for supporting our hypothesis. Below are the detailed replies.

1) For Fig 1, there are many references reported or illustrated the location of the supported Mo species on the zeolite. It is suggested that the location of Mo species and reaction path way must base on references or the nature of your study.

Response: Thank you for the reviewer's constructional comment.

The scheme of the catalyst structure and the location of the Mo species in Figure 1 are based on our characterization on the sites of Mo in zeolite and some well-recognized references¹⁻⁴.

Here we elucidate more to clarify how we rationally consider the catalyst structure.

For the Mo-based single metal catalyst, there are some references reporting the catalyst structure¹⁻⁴. The identity and anchoring sites of the initial Mo structures in such catalysts are determined as isolated oxide species with a single Mo atom on aluminum sites in the zeolite framework (acidic proton-Mo site in cages) and on silicon sites on the zeolite external surface. During the reaction, the initial isolated Mo oxide species agglomerate and convert into Mo-carbide nanoparticles. The Mo-carbide complex in the cages is responsible for methane activation and carbon coupling while the surface Mo typically triggers coke problem.

In our work, the introduction of bimetallic Pt-Bi alloy can effectively suppress the coke on the surface Mo particles, which adds another new pathway for the reaction, as shown in Fig. 1, where (1) the activation of the first C-H bond was likely to occur on the acidic proton-Mo-C site or Mo₂C on the external surface of zeolite, which leads to dissociative adsorption of methane forming surface alkyl species (CH₃*). Moreover, CH₃* are transferred to the surface of Pt-Bi particles for subsequential scission of C-H bonds to form CH₂* and preferential C-C coupling of two CH₂ species instead of the deep dehydrogenation to improve the selectivity for C₂ products and the catalyst lifetime; (2) Pt-Bi alloy mitigates the mobility of the Mo species, possibly enhancing the retention of the Mo species inside the zeolite channels during reaction. Previous studies have demonstrated that Mo-species on the external surface predominantly form coke². In the meanwhile, we have conducted a new STEM experiment of post-test catalyst, as shown below. It shows that the density of surface Mo species (small white dots) in Pt-Bi/Mo catalyst is lower than that in Mo catalyst, which indicates that the Pt-Bi alloy can restrain the mobility of Mo sites from cage to

external surface for prolonging the catalyst lifetime. This is the one of exact mechanisms we wanted to prove for explaining the reason why our tri-metallic catalyst showed improved catalytic activity and lifetime because of more active-Mo sites in the pores of zeolite. We also prove this conclusion via the acidity measurement (Fig. 5) and BET results (Supplement Table 3).

Figure 1. HAADF -STEM images of (a) Pt-Bi/Mo/ZSM-5 and (b) Mo/ZSM-5 after reaction for 120 minutes and relative EDS images of Mo/ZSM-5.

We have modified the Figure 1 to make the proposed reaction pathways clearer, as show below.

(Main Figure 2) A new proposed reaction route for activating C-H bond and coupling of C-C bond. A–C–D conventional route: reactions occurring in the pores of zeolite where Mo oxide is located: dehydrogenation and coupling of methane to ethylene are accomplished through molybdenum, and oligomerization of ethylene is performed by BASs; A–B–D new parallel route: first activation of C-H bond occurring on the Mo_2C to form CH_3^* , and then CH_3^* transfer to the surface of Pt-Bi alloy for C-C coupling. Finally, oligomerization of ethylene is also performed by BASs.

2) *The structure of Pt-Bi bimetallic particle was unclear. Only based on the evidences in Fig 2, it is not enough to conclude that Pt-Bi bimetallic particles were encapsulated outside Pt clusters, while Mo was uniformly distributed on zeolite surface.*

Response: Thanks for your suggestion.

We agree that more characterizations are needed to clarify the structure of Pt-Bi bimetallic particle. Therefore, we have conducted in-depth TEM studies to investigate morphologic information and compared with literature for validating our conclusion.

First, the prior study has identified the structure of a nanoscale Pt₃Bi intermetallic alloy⁵. The HAADF-STEM images below reveal the atomic structure of an individual Pt–Bi particle in 0.8Bi–1Pt/ZSM-5. The EDS provides intuitive elemental distribution of Pt and Bi, where a Pt–Bi alloy shell in thickness of ~1 nm could be confirmed outside a 1 nm Pt-rich core. The lattice spacing of 0.23 nm and 0.21 nm at the shell can be indexed to be (111) and (200) planes of a typical FCC phase. In addition, the line distribution of Pt and Bi shows a Pt–Bi alloy shell encapsulating outside a Pt-rich core.

Literature result⁵: (a) The high resolution HAADF image and elemental distribution of (b) Bi, (c) Si, (e) Pt, and (f) O as revealed by EDS analysis. The core–shell configuration has been clearly demonstrated by (d) the mix-over image of Pt M and Bi M signals. (g) The extracted profile of Pt M and Bi M concentration across single Pt-Bi bimetallic nanoparticle as show inset.

We have conducted more STEM characterizations to examine the catalyst morphology. As can be seen in the figure below, EDS and line profile indicate that the structure of Pt-Bi is an intermetallic alloy.

Figure 2. STEM characterizations of catalysts used in this work: (a) images and elemental distribution of (b) Mo, (c) Bi, (d) Pt, (f) Si, (g) Al and (h) Pt+Bi, as revealed by EDS analysis; (i) line distribution of Pt and Bi concentration across single Pt-Bi particle.

3) *It is unbelievable that the authors only gave the fitted curve of XPS without experimental data in Fig 2 e and f.*

Response: Thanks for the reviewer’s suggestion.

We have revised the XPS patterns carefully as shown in figure below to correctly describe the evolution of the oxidation states. From the result, we can clearly see that the shifting of binding energy for Bi 4f and Pt 4f in the single Pt, Bi, and Pt-Bi alloy in Figure 3a-b (and Figure S2) to

clearly inspect that the binding energy for Bi 4f in the alloy has shifted to a lower value compared to that in Bi–ZSM-5. In contrast, the binding energy for Pt 4f in the alloy has shifted to a higher value relatively to Pt–ZSM-5. These changes indicate that some electrons are transferred from Pt to Bi atoms in the alloy structure and therefore there is a strong electronic interaction between the Pt and Bi. The electronic perturbation of Pt by Bi is called the ligand effect.

Figure 3. XPS spectra of (a) Pt and (b) Bi in the Pt-Bi/ZSM-5 sample.

We have made the changes on the figure and revised the manuscript to include the related discussion.

4) *From Fig 2d, the Pt-Bi particle size is as large as tens of nm, which is larger than that of any scheme (Fig 1 and Fig 6, where Pt-Bi particle is smaller than pore size of ZSM-5) in this manuscript.*

Response: We would like to thank the reviewer for pointing this out.

We agree with the reviewer that the Pt-Bt particle size is tens of nanometer, and these particles are on the surface of zeolite surface, as shown in Fig. 2D. Moreover, the size of Pt-Bi particle is about 30 nm larger than pore size of ten-membered-ring (10MR) zeolite, such as MFI (ZSM-5).

The schematic figures in Fig 1 serve only the purpose of illustrating the synergistic interaction with Mo to form a sequent reaction chain and showing the anchoring sites of Mo and Pt-Bi particles. And the schematic figures in Fig 5 show only the variation of the Mo location in the zeolite.

We have added the related changes into the manuscript. (Page 3 & 9)

5) *There are no characterization results of the spent catalysts to support the reaction performance.*

Response: Thanks for your suggestion.

We have conducted the characterizations of the spent catalysts to compare the coke amount of different catalyst samples by using TGA.

As shown in figure 4 below, each catalyst sample shows an initial weight loss at temperature below 250 °C attributing to the loss of humidity⁶. Weight loss between 480 and 680°C corresponds to the burn-off of coke that has deposited during reaction⁷. By comparing Mo/ZSM-5 catalysts with different loading, it is observed that the amount of deposited carbon increases with loading, i.e. from 1.0 to 2.0 wt.%. In addition, comparing the amount of burn-off coke during reaction in the Mo versus the Pt-Bi/Mo catalyst, it is evident that there is more coke formation in the former ones, which coincides with previous experimental results as illustrated in Fig. 3C. In other words, the presence of Pt-Bi alloy is beneficial for extending catalyst lifetime.

We have added the related discussion into manuscript. (Page 5)

Figure 4. (a) TGA profiles of different spent catalysts after 8h TOS at 710 °C; (b) Amount of calculated coke burn-off in spent catalyst.

6) *Higher initial conversion is obviously induced by side reaction via the added Pt, and the author should discuss it in detail.*

Response: Thank you for your suggestion.

We agree with reviewer that the initial high conversion is partially related with the added Pt. As can be seen in Fig. 3A, the presence of Pt improves the methane conversion compared with 1 wt% Mo catalyst, while methane is more likely to form coke based on the carbon balance of product selectivity. Meanwhile, C₂H₆-TPD profile (Fig. 6D) showed that the desorption temperature of Pt/Mo/ZSM-5 is the highest and the peak is wide, which indicates that Pt promoting further dehydrogenation of C₂H₆ on Mo₂C sites, further reveals Pt enhances deep dehydrogenation of methane (forming CH₂, CH and C species). In contrast, the catalysts activity was stabilized after an initial transient activation period (~90 min). When we compared methane conversion between

Mo/ZSM-5 and Pt-Bi/Mo/ZSM-5 catalysts, higher conversion is mainly caused by the synergetic effect between Pt-Bi and Mo. Like our DFT results show, the energy barrier for the formation of CH_3^* in the Pt-Bi/Mo catalyst is lower than that in the Mo catalyst and the subsequent C_2H_4 formation is more competitive.

We have added the related discussion into manuscript. (Page 5 & 10 & 13)

Reviewer # 2

Comments to the Author

This study reports an efficient trimetallic catalyst system (Pt-Bi-Mo) on HZSM-5 to selectively convert methane to aromatics. The authors showed that adding Pt-Bi species to Mo/HZSM-5 enhanced both methane conversion and aromatic selectivity, and that its active sites were reversibly regenerated through periodic reactions and H_2 regeneration.

1) However, the synergy between Pt-Bi and Mo remains unclear. The authors should note that the MDA performance of the Pt-Bi-Mo catalyst is not the same as that of the parallel activities of the Pt-Bi and Mo catalysts even though the addition of Bi to Pt has a positive effect on the monometallic Pt catalyst.

Response: We thank the reviewer for the valuable suggestions.

We understand that the synergy between Pt-Bi and Mo is the most important mechanism for explaining the results. Together with other reviewer's comment, we have performed DFT study and more characterizations to further investigate the underlying mechanism.

Since the Reviewer #3 also gave the suggestion and it is a long reply, we appreciate the reviewer referring "**reply to Reviewer #3**" about elucidating the synergistic interaction between Pt-Bi alloy and Mo species.

2) Since the literature reported that the optimal Mo content on HZSM-5 is above 5 wt%, the authors need to compare the activity with the optimized 5wt% MoO_3 /HZSM-5 catalyst, which should be performed the MDA reaction at same condition.

Response: Thank you for your suggestion.

We have several considerations on selecting a total 2% metal loading: (1) we focused on understanding the promotional effect of Pt-Bi on improving C_2 coupling and suppressing coke formation, and the optimization of Mo content was not our target for this new tri-metallic catalyst system; (2) the lower metal loading would facilitate the identification of promotional effects from addition of Pt-Bi. If 5% was adopted, more Pt-Bi percentage is required to observe the synergistic effect.

We have measured the performance of 5wt% Mo/ZSM-5 catalyst and added the result into Fig. 3E. The result shows that benzene yield of 5%Mo catalyst is about 20% higher than 2%Mo catalyst but is ~18% lower than Pt-Bi/Mo catalyst. Since the composition would vary a lot for this study,

it will be in our future work scope that Mo content and relative ratio of Mo to Pt/Bi will be further optimized and expected to be very interesting work.

3) *Since the XRD analysis of the catalyst used cannot fully prove that coke was not formed, it is necessary to present a carbon balance of product selectivity. Additionally, it is a good way to analyze the coke deposited on spent catalysts through TOC or TGA analysis.*

Response: Thank you for your suggestion.

We agree with reviewer's opinion that TGA is a more effective approach to validate the absence of coke. Therefore, we have performed TGA analysis of the tested catalysts.

As shown in Figure 4, each catalyst sample shows an initial weight loss at temperature below 250 °C attributed to the loss of humidity. Weight loss between 480 and 680°C corresponds to the burn-off of coke that has deposited during reaction. By comparing Mo/ZSM-5 catalysts with different loading, it is observed that the amount of deposited carbon increases with loading, i.e. from 1.0 to 2.0 wt.%. In addition, comparing the amount of burn-off coke during reaction in the Mo versus the Pt-Bi/Mo catalyst, our results can mainly conclude that less coke on the tri-metallic catalyst was found than that on the Mo monometallic catalyst, which coincides with previous experimental results as illustrated in Fig. 3C. In other words, the presence of Pt-Bi alloy is beneficial for extending catalyst lifetime.

4) *The authors suggested that the synergistic effect of Pt-Bi on Mo/HZSM-5 was due to increased methane adsorption and reduction through characterizations. However, it could be argued that the decrease in the apparent activation energy of the Pt-Bi-Mo catalyst enhanced C-H activation compared to that of the related Mo catalyst; however, it may also promote coke formation. In fact, the activation energy of the Pt-Bi-Mo catalyst was determined to be similar to the energy required to form coke from methane. Therefore, a detailed discussion should be made based on absolute coke selectivity.*

Response: Thank you for your comment.

It is very interesting to discuss this question because this trimetallic catalyst can enhance the MDA reaction activity and relatively suppress the coke.

Overall, from the perspective of enhancing the activity, the synergistic effect of Pt-Bi on Mo/ZSM-5 is that the formed CH_3^* from Mo_2C on the external surface of zeolite can efficiently move on for C-C coupling on the surface of Pt-Bi particle to produce C_2 compounds, which are the key intermediates of oligomerization. This new pathway is parallel with the catalysis on Mo inside the cage.

First, we compared the catalytic activity between Mo and Pt-Bi/Mo catalysts based on absolute selectivity as shown in Fig. 3B-C. The results showed that Pt-Bi/Mo catalyst had higher methane conversion and benzene selectivity while lower coke selectivity.

Second, we performed the measurement of TGA on tested catalysts to study the effect of composition on coke. It is found that the Pt-Bi-Mo/ZSM-5 catalyst showed the less coke than

others, which is consistent with the stability test and regeneration result. We think the suppressed coking is related with the change of surface catalyst distribution. Typically, the carbon deposition is caused by the continuous and uncontrollable C-H bond cleavage on Mo-based catalyst, which is regarded as a common issue for this type of catalyst. By adding Pt-Bi, the activated methane forms the intermediates that can quickly transfer to Pt-Bi surface for C-C coupling instead of coke. At the same time, we found that Pt-Bi can reduce the migration of Mo from cage to outer surface, as illustrated in Fig. 5.

Therefore, this catalyst can relatively avoid the fast degradation and can recover quickly from the regeneration process.

The related discussion has been added into the manuscript. (Page 3 & 5 & 8)

Reviewer #3 (Remarks to the Author):

Dear Authors,

Methane dehydroaromatization (MDA) under non-oxidative conditions is an interesting route for upgrading methane. Despite the thermodynamic limitation of methane conversion and the fast deactivation of zeolite-based catalysts (usually Mo/ZSM-5), the process is highly selective to benzene and H₂ is obtained as by-product. Therefore, it has been thoroughly studied along the last 3 decades. Although much has been published regarding the nature of the active sites, formed under working conditions, the reaction mechanism (mono- or bi-functional) and the reaction intermediates (acetylene, ethylene, surface-formate, acetal, allene...), there is still a strong debate on all these issues. Thus, the topic of this manuscript (MS), which addresses the improvement of the classical Mo/ZSM-5 catalyst by incorporation of Pt-Bi alloys, is of general interest and would be suitable for Nature Communication. The results obtained evidence important consequences derived from the incorporation of Pt-Bi to the Mo/ZSM-5 catalyst, such as the increased reducibility of the Mo species or the increase in methane conversion, benzene selectivity and catalyst life.

However, in my opinion, some fundamental questions remain open, such as the role of the Pt-Bi alloy and the Mo species in the activity/selectivity, the reason for the higher catalyst life and benzene selectivity and the role of Pt-Bi on the migration of Mo to the catalyst surface, and the work does not meet the quality standards for publication in this journal.

Response: We would like to thank the reviewer's comment and constructive suggestions on improving this manuscript. We agree that the synergy between Pt-Bi and Mo needs to be explained for understanding the effects on activity, lifetime, and suppression of Mo migration. Therefore, we have conducted more experimental work and DFT study to elucidate the fundamental mechanisms. Please refer to the section below for more specific details.

I suggest its submission to a more specialized journal after major revisions. Moreover, in the way the manuscript is organized, a lot of doubts and question come to the mind of the reader, which may find an answer further in the article, but make it very difficult to read and to follow.

Response: We would like to thank the reviewer's comment. After receiving the comments from three reviewers, we have made a major revision on manuscript to improve the fundamental understanding on the catalysis by **1) adding more experimental results; 2) carrying out DFT investigation; and 3) modifying and re-organizing the figures and the manuscript.** Therefore, the authors believe that the current version is suitable for further consideration in this journal for publication.

In addition, a native-speaking senior scientist in Idaho National Laboratory was invited to improve the language for high readability.

Finally, the Discussion section is more likely a Conclusion section.

Response: Thanks for your suggestion. We have revised the section to have more discussions on our characterizations and DFT results to elucidate the synergy between Pt-Bi and Mo and improvement of catalysis stability.

DFT study for understanding the synergy between Pt-Bi and Mo

In our work, we hypothesize that the Pt-Bi bimetallic alloy plays a key role in **a) facilitating the C₂ species formation rate and b) stabilizing the catalyst performance by suppressing the movement of Mo from cage to surface.** The Mo species and Pt-Bi function differently in the reactions.

First, based on the shape-selective nature of benzene formation and spectroscopic observations, the Mo species are most likely located inside the zeolite pores². The proximity of the acidic proton-Mo sites is a key factor in determining the catalyst lifetime as the migration of Mo species to the zeolite external surface leads to quick catalyst deactivation⁸. **The activation of the C-H bond occurs on the acidic proton-Mo-C site/ Mo₂C on the external surface of zeolite,** which leads to dissociative adsorption of methane forming surface alkyl species (CH₃^{*}).

Second, **CH₃^{*} are transferred to the surface of Pt-Bi particles for subsequential scission of C-H bonds to form CH₂^{*} and preferential C-C coupling of two CH₂ species instead of the deep dehydrogenation to improve the selectivity for C₂ products and the catalyst lifetime;** Third, the Pt-Bi particle suppresses the mobility of Mo species during the reaction to improve coke resistance.

The figure below shows the general processes of methane to C₂ species.

Literature result⁹: Formation of ethylene through methane activation following three different schemes.

Figure 5. Construction of catalyst models. Relaxed bulk Mo_2C and Pt-Bi crystals are cleaved to expose their (001) and (111) facets, respectively. The PtBi-MoC composite was built by placing a periodic 1D Mo_2C nanowire on Pt-Bi(111) to represent the Mo_2C and Pt-Bi co-catalyst. Color scheme: Mo (purple), C (brown), Bi (magenta), and Pt (grey).

In our revision work, **we have conducted DFT study to find the synergy between Pt-Bi alloy and Mo species and to understand each role in catalysis.** Based on the assumption that the initial conversion of CH_4 into the precursor of benzene (i.e., C_2H_4) takes place at the metallic sites, the primary active site configurations for the CH_4 -to- C_2H_4 conversion were constructed using Mo_2C and Pt-Bi bulk crystals. The structure of Mo_2C (mp-1552), with the lowest formation energy, was extracted from the Materials Project. The Pt-Bi alloy was constructed using a $(2 \times 2 \times 2)$ face-centered cubic Pt supercell (see Figure 5). Ionic relaxations of these bulk structures were carried out using a cut-off energy of 520 eV and an automatically generated $6 \times 6 \times 6$ k -point mesh based on the Monkhorst-Pack scheme¹⁰. The convergence criterion for the self-consistent energy loop is set to be 1×10^{-6} eV, and the ionic steps stop when the forces on each relaxed atom are less than 0.02 eV/Å. The orbital partial occupancy was determined using the Methfessel-Paxton method¹¹. The lattice constants are relaxed to $a = 4.74$ Å, $b = 5.23$ Å, and $c = 6.06$ Å for the orthorhombic Mo_2C , which compare well with literature values¹². The lattice constants for Pt-Bi become a =

8.12 Å and $b = c = 8.14$ Å. Due to the large Bi atomic radius, the cell expands by approximately 2.3% compared to pure Pt bulk ($a = b = c = 7.95$ Å). Lattice strain has a pronounced effect on metal surfaces. Nørksov and coworkers^{13,14} showed that the binding strengths of surface adsorbates increase and molecular activations (e.g., CH₄, CO, H₂) become easier as the *d*-band center shifts to a higher energy level when the lattice expands. Hence, the effect of Pt-Bi alloy on the CH₄-to-C₂H₄ conversion will be understood by examining its elementary steps.

Surface reaction steps depicting a simplified CH₄-to-C₂H₄ conversion process are modeled using DFT calculations on Mo₂C semi-periodic Pt₃Bi employing Mo₂C(001) and Pt-Bi(111) slabs (see Figure 5). A cutoff energy of 400 eV and a $4 \times 4 \times 1$ *k*-point mesh were used, while other parameters remained unchanged. Moreover, a composite model derived from Mo₂C and Pt-Bi was employed to investigate a possible synergistic pathway facilitating the CH₄-to-C₂H₄ conversion in the co-catalyst setting. A one-dimensional (1D) stoichiometric Mo₂C wire was placed on top of the Pt-Bi (111) surface (denoted as PtBi-MoC in Figure 5). This configuration was chosen because of the similarity of lattice parameters between Mo₂C and Pt-Bi. The *front* side of the wire was used to study surface chemistry.

Figure 6. Conversion of CH₄ into C₂H₄ on Pt-Bi(111) (red dashed lines), Mo₂C(001) (black dashed lines), and PtBi-MoC (solid green lines) surfaces. The free energy changes for CH₄ dissociation, H₂ and C₂H₄ desorptions are estimated at 700 °C and 1 bar. The configurations of adsorbed CH₄, CH₃, CH₂, C₂H₄, and H are displayed as inset figures.

All surface intermediates except CH₄, that is, CH₃, CH₂, C₂H₄, and H bind more strongly on the Mo₂C(001) than on Pt-Bi(111) (see Table 1), thus, making the initial CH₄ activation and conversion energetically favorable. These results agree with past studies in the literature, where the Mo/ZSM-5 is a known effective catalyst for the methane dehydroaromatization process. Moreover, our experiments also suggested that Pt-Bi does not display notable reactivity toward methane conversions. A closer examination indicates that **the Mo species is responsible for the strong interactions between H and carbon (CH_x and C₂H₄) species. In the Pt-Bi alloy, H and carbon species prefer to bind with the Pt species. Also, CH₃ and CH₂ favor the site that satisfies the electronic valency of the central carbon atom, i.e., CH₃ prefers the Pt top site, while CH₂ is likely located between two Pt atoms.**

Table 1. Binding energies (in eV) and site preference on Pt-Bi(111) and Mo₂C(001).

Species	Pt-Bi(111)		Mo ₂ C(001)	
	BE (eV)	Site preference	BE (eV)	Site preference
CH ₄	-0.02	Bi top	-0.02	Mo top
CH ₃	0.32	Pt top	-0.12	Mo-Mo bridge
CH ₂	0.91	Pt-Pt bridge	0.26	Mo-Mo bridge
C ₂ H ₄	-1.33	Pt-Pt bridge	-1.97	Mo-Mo bridge
H	-0.52	3-fold fcc Pt	-0.92	Mo-Mo bridge

As shown in Figure 6, the activation of the first CH₄ C–H bond is highly endothermic on all surfaces due to entropy loss, with an energy barrier of 0.5~0.6 eV. Because of the strong binding of CH₃ and H on Mo₂C(001), the reaction is much less endergonic than on Pt-Bi(111) (by ~0.9 eV). The subsequent decomposition of CH₃ is distinct between the two surfaces, where **CH₃-to-CH₂ becomes exothermic (-0.54 eV) on Mo₂C(001) but remains slightly endothermic (0.14 eV) on Pt-Bi(111)**. Afterward, the formations of gas-phase H₂ and adsorbed C₂H₄ via recombinations of H and CH₂ on Mo₂C(111) become endothermic. In contrast, **these steps are much more thermodynamically favorable on Pt-Bi(111)**, as depicted by the downward shifting free energy landscape.

Although the final C₂H₄ desorption step is exergonic on both surfaces, the free energy surfaces displayed in Figure 6 describe very different pathways on separated Mo₂C and Pt-Bi catalysts. The initial activation and conversion of CH₄ on Mo₂C(001) benefits from the strong bindings with the Mo sites with respect to the prohibitive thermodynamic barrier on Pt-Bi. Nevertheless, once CH₄ is activated, both H₂ and C₂H₄ formations are competitive thanks to the moderate binding with the Pt site. **When Mo₂C and Pt-Bi alloy are combined, the CH₄-to-C₂H₄ process can take advantage of such distinct catalyst behaviors.** Based on the Pt-Bi-MoC composite model (Figure 5), the initial CH₄ activation pathway was determined using the CI-NEB method. We deliberately chose the supported Mo₂C front edge owing to the demonstrated reactivity for CH₄ conversions. Such edge site shall mimic the interface established by the two primary catalytic materials due to its proximity to the Pt-Bi underneath. An energy barrier of approximately 0.55 eV was obtained (Figure 7), which is in line with that on Mo₂C(001). As discussed previously, CH₄ activation is mainly hindered by the thermodynamics of this elementary step. We noted that **the reaction free energy is reduced by approximately 0.27 eV, lower than both PtBi(111) and Mo₂C(001)**. Such a reduction benefits from the edge Mo sites that anchor and stabilize the CH₃ group upon the C–H bond activation, as evidenced by the potential energy valley shown in Figure 7. We also observed that, **for CH₃ to migrate across the interfacial boundary onto Pt-Bi, a low energy barrier of ~0.4 eV is needed.** Under the MDA temperature, such migration would be facile. On the Pt-Bi support, the conversion of CH₃ into CH₂ is endothermic (0.40 eV). The subsequent C₂H₄ formation will follow a similar path to that on Pt-Bi(111). With the composite model, the overall free energy landscape has acquired the behaviors from both its components. The initial CH₄ activation resembles that of Mo₂C(111). Hence, PtBi-MoC not only retains, but also somewhat improves the reactivity. As the reaction proceeds, **carbonaceous species migrating onto Pt-Bi will more likely continue to form C₂H₄, which eventually desorbs from the system easily instead of occupying and blocking the active sites.**

Figure 7. Potential energy surface for CH₄ activation at the Mo₂C-Pt-Bi interface.

The DFT discussion above has been added into manuscript and supporting material to reflect the changes. (starting from Page 12)

Additional comments are listed below:

1. *The title is not appropriate and could be misleading, as MDA is not a carbon-free process. Although the carbon footprint is lower as compared to oil-based processes, it uses a fossil fuel as carbon source, the thermochemical process takes place at high temperatures (energy intensive and corresponding CO₂ emissions) and coke is formed as a by-product.*

Response: We thank the reviewer for the good suggestion. Claiming a carbon-free process is not proper. Therefore, we have changed the title to “Direct conversion of methane to aromatics and hydrogen via a heterogeneous trimetallic synergistic catalyst”.

2. *The literature cited in the MS is incomplete. No references are made to publications from the group of Hensen and Kosinov or Gascon, highly active in the field. The description of the reaction mechanism and the reaction intermediates is completely outdated and in some aspects, incorrect. For instance, ethane is formed during the process but cannot be considered an intermediate because it is a stable product that will not react any further under MDA reaction conditions.*

Response: We thank the reviewer for the suggestion.

We have added more literature from the suggested research groups and other teams in both introduction and discussion parts.

A senior scientist in Idaho National Laboratory with expertise in catalysis was invited to revise our manuscript to eliminate any unprofessional terms or statements.

Below is a paragraph as an example of describing the mechanism:

“In this process, dehydrogenation and oligomerization of methane occur on active Mo sites forming C₂ intermediates such as ethene and acetylene, followed by cyclization producing aromatics and naphthalene on the Brønsted acid sites (BASs) in zeolite⁸. It is generally accepted that partially reduced and/or carburized Mo-oxo species in zeolite channels such as Mo₂C and oxycarbide Mo (MoO_xC_y) serve as the active sites while the Mo carbides on the external surface are less active, and the proximity of the acidic proton-Mo sites is shown to be a key factor in determining the catalyst lifetime as the migration of Mo species to the zeolite external surface leads to quick catalyst deactivation⁸.”

3. How and why does Mo migrates to the surface in the presence of the Pt-Bi alloy? Which is the role of Pt-Bi in this migration? No explanation is given for this although the authors present some experimental and theoretical evidences, such as the acidity measurements of the simulations presented in Figure 6. This part of the discussion should be improved. Moreover, if Pt-Bi activate the methane’s C-H bond, which is the role of the Mo species?

Response: Thanks for the reviewer’s comment.

It is very interesting to observe the bi-functional roles of Pt-Bi on the catalyst performance: (a) the synergy between Pt-Bi and Mo promoting the catalysis activity and (b) the impact of Pt-Bi on Mo migration.

The mechanism leading to Mo migration is not very clear so far. However, it has been discovered that the Mo-carbide nanoparticles with a C/Mo ratio greater than 1.5 are more stable on an external Si site of H-ZSM5 than they are on an Al site, which provides the driving force for the migration of Mo-carbide from internal cages onto the external surface of the zeolite¹. Therefore, the Pt-Bi possibly affects the migration energy or pathway. We will pursue along this direction in our future work.

For reviewer’s second question, it is considered that the surface Mo species dominate the methane activation and Pt-Bi facilitates the C₂ coupling on zeolite surface although Pt-Bi is partially involved into methane activation reaction. To avoid any confusion, we have clarified this discussion in the manuscript. (Page 3 & 11)

4. How does the Pt-Bi alloy evolves when exposed to the reaction-regeneration cycles? And the Mo species? Why do the authors use 1 wt% Pt? Pt loading is not optimized.

Response: Thank you for your comment.

We have performed the morphology examination before and after the test, as shown below:

Figure 8. HAADF -STEM images of (a) before and (b) after test and XRD profile (c) of Pt-Bi/Mo catalyst before and after the reaction.

From the catalyst surface, there is no visible changes observed after the regeneration test. Furthermore, the TGA analysis indicates that the coke on Pt-Bi/Mo catalyst is greatly suppressed.

The selection of 1% Pt loading was based on the prior work that 1 wt% Pt-Bi/ZSM-5 catalyst has high selectivity of C_2 species for nonoxidative coupling of methane. Because we are investigating the synergy between Pt-Bi and Mo, the Pt loading has not been optimized in this work. However, it would be an interesting topic for our future work.

5. How do the C_2H_6 -TPD results proof that the Pt-Bi/Mo/ZSM-5 catalyst yields more C_2 ? This part of the discussion should be improved.

Response: Thank you for your suggestion.

The TPD results can accurately measure the desorption behaviors with dependence on the temperature. In this work, we used C_2H_6 -TPD results to explain that the different desorption temperature on catalyst surface indicates the activity of C_2 intermediates for further aromatization reaction.

The higher desorption temperature indicates a stronger bonding between the adsorbate species and surface, which leads to deep dehydrogenation. Fig. 6D showed that the desorption temperature of Pt/Mo/ZSM-5 is the highest and the peak is wide, which indicates that Pt promotes further dehydrogenation of C_2H_6 on Mo_2C sites. Therefore, the addition of Pt enhances deep dehydrogenation of methane (forming CH_2 , CH and C species).

In contrast, the desorption temperature ($291^\circ C$) on Pt-Bi/Mo/ZSM-5 catalyst is lower than that of Mo/ZSM-5 ($345^\circ C$), which was ascribed to that Pt-Bi particle facilitates the desorption of surface alkyl species on Mo-C site of the zeolite surface to form methyl radicals. These CH_3^* are further transferred to the surface of Pt-Bi particles for preferential C-C coupling instead of continuous cleavage of C-H bonds to improve the selectivity for C_2 products.

We have added the relevant discussion to the manuscript.

6. *Atomically dispersed Mo species in the 10-ring channels are hardly distinguished in Figure 2c.*

Response: Thank you for the comment.

We agree with reviewer that it is hard to distinguish the morphology of dispersed Mo species in the 10-ring channels in Fig. 2.

Below is our target image from a group using integrated differential phase-contrast scanning transmission electron microscopy (iDPC-STEM) to illustrate the isolated nature of Mo-oxo complexes inside the zeolite pores in 4 wt% Mo/ZSM-5 catalyst.

Literature result¹⁶: iDPC-STEM image of Mo/ZSM-5, showing the presence of off-center contrast in many 10-MR channels. Insets show zoomed-in areas 1, 2, and 3 respectively and corresponding intensity line profiles.

In the past efforts, we have our samples characterized in several different institutions and however, the zeolite structure under TEM is not strong enough to resist electron beam and the shifting image seriously affects the observation.

After receiving the revision decision, we sent our samples to another university for another try, and we received some better images for element distribution of the catalysts, as shown in Fig. xx, but missed any images of the Mo at 10-ring channels.

7. *Page 4, lines 108-109: According to the text, the ligand effect make Pt more “atomic like” binding surface species more weakly, further enhancing the catalytic performance. Which surface species? Binding with the support?*

Response: Thank you for the comment.

The ligand effect makes Pt binding surface alkyl species (CH_3^*) more weakly to form methyl radicals, which are subsequent C-H activation to form CH_2^* and preferential $\text{CH}_2\text{-CH}_2$ coupling instead of scission of the remaining C-H bonds. It is known that ethylene is an important intermediate for the aromatics formation, further, the yield of benzene was enhanced.

We have revised the manuscript to reflect the changes. (Page 4)

8. Page 4, line 110: which is the assignment of the XRD peak at $2\theta = 39.9^\circ$ that is observed for the Pt-Bi-ZSM-5 samples?

Response: Thank you for the comment.

The weak peak at $2\theta = 39.9^\circ$ corresponds to Pt-Bi alloy particle. This can further prove that the Pt-Bi particles are well dispersed on the support surface.

9. Page 5, lines 157-159: the authors conclude that there is no coke on the catalyst based on XRD measurements of the used ZSM-5. However, some deactivation is observed. Thermogravimetric analysis or elemental analysis would confirm the degree of coke on catalyst.

Response: Thank you for your suggestion.

As reviewer suggested, we have performed TGA measurement to affirm the amount of carbon deposited in the Pt-Bi/Mo catalysts during reaction. As shown in Figure 4, each catalyst sample shows an initial weight loss at temperature below 250°C attributed to the loss of humidity. Weight loss between 480 and 680°C corresponds to the burn-off of coke that has deposited during reaction. By comparing Mo/ZSM-5 catalysts with different loading, it is observed that the amount of deposited carbon increases with loading, i.e. from 1.0 to 2.0 wt.%. In addition, comparing the amount of burn-off coke during reaction in the Mo versus the Pt-Bi/Mo catalyst, our results can mainly conclude that less coke on the tri-metallic catalyst was found than that on the Mo monometallic catalyst, which coincides with previous experimental results as illustrated in Fig. 3C. In other words, the presence of Pt-Bi alloy is beneficial for extending catalyst lifetime.

The changes are integrated into the manuscript. (Page 5)

10. Feed gas is 95% methane. Which gas completes the mixture? Please specify.

Response: We used 95%CH₄/5%N₂ for feeding gas. This has been added into manuscript. (Page 1)

11. Do the authors use an internal standard for carbon mass balance determination? Nothing is mentioned in the Experimental Section.

Response: Thank you for your suggestion.

We have added more details in the Experimental Section: (Page 15)

“The catalysts were tested in a fixed-bed tubular quartz reactor with 7 mm inner diameter and a length of 450 mm. The catalyst bed consisted of 500 mg of sample, held in place between two quartz wool plugs. The samples were heated to 710°C with a $5^\circ\text{C}/\text{min}$ temperature ramp, under Ar gas with flow rates of 20 sccm; after temperature equilibration at 700°C was achieved, reaction

conditions were applied by flowing 95 % CH₄/N₂ into the reactor at atmospheric pressure and at a space velocity of 1272 mL·gcat⁻¹·h⁻¹. With nitrogen as an internal standard, the products leaving the reactor were analyzed using a gas chromatograph-mass spectrometer equipped with flame ionization detector and high-performance ion source (Shimadzu GCMS-QP2020NX).”

12. Figures are too small, difficult to read. Figure 3e should be updated with more recent results from groups experts in the field such as the one from Prof. Hensen or Prof. Gascon. The same applies to Table 1 in the SI.

Response: Thank you for your suggestion.

We have optimized all figures to make sure every detail can be read. As reviewer suggested, more results from these groups have been added to our comparisons.

References

- 1 Gao, J. *et al.* Identification of molybdenum oxide nanostructures on zeolites for natural gas conversion. *Science* **348**, 686-690 (2015). <https://doi.org:10.1126/science.aaa7048>
- 2 Kosinov, N. *et al.* Methane Dehydroaromatization by Mo/HZSM-5: Mono- or Bifunctional Catalysis? *ACS Catalysis* **7**, 520-529 (2017). <https://doi.org:10.1021/acscatal.6b02497>
- 3 Kosinov, N. *et al.* Stable Mo/HZSM-5 methane dehydroaromatization catalysts optimized for high-temperature calcination-regeneration. *Journal of Catalysis* **346**, 125-133 (2017). <https://doi.org:https://doi.org/10.1016/j.jcat.2016.12.006>
- 4 Vollmer, I. *et al.* Relevance of the Mo-precursor state in H-ZSM-5 for methane dehydroaromatization. *Catalysis Science & Technology* **8**, 916-922 (2018). <https://doi.org:10.1039/C7CY01789H>
- 5 Zhu Chen, J. *et al.* Identification of the structure of the Bi promoted Pt non-oxidative coupling of methane catalyst: a nanoscale Pt₃Bi intermetallic alloy. *Catalysis Science & Technology* **9**, 1349-1356 (2019). <https://doi.org:10.1039/C8CY02171F>
- 6 Jiang, S., Zhang, H., Yan, Y. & Zhang, X. Stability and deactivation of Fe-ZSM-5 zeolite catalyst for catalytic wet peroxide oxidation of phenol in a membrane reactor. *RSC Advances* **5**, 41269-41277 (2015). <https://doi.org:10.1039/C5RA05039A>
- 7 Liu, B. *et al.* Methanol-to-hydrocarbons conversion over MoO₃/H-ZSM-5 catalysts prepared via lower temperature calcination: a route to tailor the distribution and evolution of promoter Mo species, and their corresponding catalytic properties††Electronic supplementary information (ESI) available: more TEM images of post-run samples, CS Chem3D Model of zeolite and external surface MoO₃, images and file (.c3xml). See DOI: 10.1039/c5sc01825k. *Chemical Science* **6**, 5152-5163 (2015). <https://doi.org:https://doi.org/10.1039/c5sc01825k>
- 8 Gao, W. *et al.* Dual Active Sites on Molybdenum/ZSM-5 Catalyst for Methane Dehydroaromatization: Insights from Solid-State NMR Spectroscopy. *Angewandte Chemie International Edition* **60**, 10709-10715 (2021). <https://doi.org:https://doi.org/10.1002/anie.202017074>
- 9 Khan, T. S., Balyan, S., Mishra, S., Pant, K. K. & Haider, M. A. Mechanistic Insights into the Activity of Mo-Carbide Clusters for Methane Dehydrogenation and Carbon-

- Carbon Coupling Reactions To Form Ethylene in Methane Dehydroaromatization. *The Journal of Physical Chemistry C* **122**, 11754-11764 (2018).
<https://doi.org:10.1021/acs.jpcc.7b09275>
- 10 Monkhorst, H. J. & Pack, J. D. Special points for Brillouin-zone integrations. *Phys. Rev. B* **13**, 5188-5192 (1976). <https://doi.org:10.1103/PhysRevB.13.5188>
- 11 Methfessel, M. & Paxton, A. T. High-precision sampling for Brillouin-zone integration in metals. *Physical Review B* **40**, 3616-3621 (1989).
<https://doi.org:10.1103/PhysRevB.40.3616>
- 12 Naher, M. I. & Naqib, S. H. Possible applications of Mo₂C in the orthorhombic and hexagonal phases explored via ab-initio investigations of elastic, bonding, optoelectronic and thermophysical properties. *Results in Physics* **37**, 105505 (2022).
<https://doi.org:https://doi.org/10.1016/j.rinp.2022.105505>
- 13 Mavrikakis, M., Hammer, B. & Nørskov, J. K. Effect of Strain on the Reactivity of Metal Surfaces. *Physical Review Letters* **81**, 2819-2822 (1998).
<https://doi.org:10.1103/PhysRevLett.81.2819>
- 14 Abild-Pedersen, F., Greeley, J. & Nørskov, J. K. Understanding the Effect of Steps, Strain, Poisons, and Alloying: Methane Activation on Ni Surfaces. *Catalysis Letters* **105**, 9-13 (2005). <https://doi.org:10.1007/s10562-005-7998-9>
- 15 Rakić, V. & Damjanović, L. Temperature-programmed desorption (TPD) methods. *Calorimetry and thermal methods in catalysis*, 131-174 (2013).
- 16 Wang, N. *et al.* Probing the Catalytic Active Sites of Mo/HZSM-5 and Their Deactivation during Methane Dehydroaromatization. *Cell Reports Physical Science* **2**, 100309 (2021). <https://doi.org:https://doi.org/10.1016/j.xcrp.2020.100309>

REVIEWER COMMENTS

Reviewer #2 (Remarks to the Author):

The authors have revised all the comments made by the previous reviewers, and I think the paper deserves to be published in Nature Communications.

Reviewer #3 (Remarks to the Author):

I recommend the manuscript to be ACCEPTED after major revision.

The authors have significantly improved the manuscript by addressing the suggestions and comments of the different reviewers. Moreover, the new HAADF-STEM study helps understanding structure of the Pt-Bi alloy at the atomic level, and the DFT study provides a coherent explanation for the synergy observed between the Mo and the Pt-Bi species. The kinetic study and the activation energies obtained also evidence the benefits of adding the Pt-Bi alloy.

However, in my opinion the work should still be improved in order to meet the quality standards required for publication in Nature Communications.

Details are listed below:

1. Page 2, lines 64-66: the authors mention three types of coke, but only specify two, soft and hard coke.
2. Figure 1: although the reaction pathway is presented in Figure 1, I think it would be useful to have the individual elementary steps of the process proposed in the text as well.
3. Page 3, lines 83-85: At some points of the manuscript, such as this one, it is confusing when the authors refer to which of the active species are on the external surface of the zeolite crystals or inside of the structure on the micropore surface. For instance, here, in line 83, the authors should specify they are referring to the surface within the pores.
4. Page 3, lines 88-90: Although some results may be anticipated in the Introduction, I think there is no reason for referring to a particular figure (Figure S1).
5. Page 4, lines 111-112: according to the authors "Mo particle is 3-5 nm that less than pore size of ten-membered-ring (10MR) zeolite, such as MFI (ZSM-5)." This sentence should be rephrased.

Moreover, a particle of 3-5 nm in size will not fit within the pores of the MFI structure, with sizes in the range of 0.5-0.55 nm. In fact, nothing can be distinguished in Figure 2-C.

6. Page 5, line 151: Reference is made to Figures S4-S9. In these graphs, please use the same range in the scale for CH₄ conversion (0-25%).

7. Page 5, line 173: the authors discuss the "conversion rates". However, the results they give are no rates but methane conversion degrees (%). This should be corrected along the whole manuscript.

8. Page 6, lines 179-188: the authors discuss the reduction of coke formation when adding Pt-Bi, and relate it with "a suppressed scission of C-H bonds on the external Mo species and with the reduced mobility of acidic proton-Mo in the pore of zeolite." Did the authors consider the possible contribution of coke precursors' hydrogenation due to presence of Pt?

9. Figure 3A-C: please specify the TOS at which the results are compared.

10. Page 7, lines 230-232: Please rephrase, the sentence is difficult to understand. Which previous study do the authors refer to?

11. Table S1: Please check the space velocity given for Mo catalyst, reference 8. Perhaps it should be 2000 and not 2.0.

12. Figures 5A-C (page 9) are illustrative of the experiments performed to study the influence of the Pt-Bi alloy on the mobility of the Mo species. Are these figures (A-C) based on theoretical calculations, or only cartoons of what the authors believe is happening when increasing Si/al ratio of the MFI or by adding Pt-Bi? Moreover, did the authors consider possible blocking of the pore mouths by the Pt-Bi particles as a possible cause for the reduced mobility of the Mo species?

13. Page 9, lines 303-304: "This result indicates the addition of the Pt promoted the reducibility of the molybdenum oxides by providing activated hydrocarbon atoms." Could the authors explain this better?

14. Page 10, discussion on the reaction kinetics: did the authors test the Pt-Bi catalyst (Mo-free)? I think it would be interesting to compare the activation energy for this catalyst with the other three in Figure 6C. It would also give information on the controlling step for the whole process. Which have been the reaction temperatures used for this study? How were the E_a obtained, at which TOS? I have not found this information in the Methods section.

15. Page 12, Figure 7. Please refer to Figure S17, in the supporting information, so the reader can understand the color codes and the models used for the DFT study.

Point-to-point responses to the reviewer' comments and suggestions

Reviewer #3 (Remarks to the Author):

Comments to the Author

I recommend the manuscript to be ACCEPTED after major revision.

The authors have significantly improved the manuscript by addressing the suggestions and comments of the different reviewers. Moreover, the new HAADF-STEM study helps understanding structure of the Pt-Bi alloy at the atomic level, and the DFT study provides a coherent explanation for the synergy observed between the Mo and the Pt-Bi species. The kinetic study and the activation energies obtained also evidence the benefits of adding the Pt-Bi alloy.

However, in my opinion the work should still be improved in order to meet the quality standards required for publication in Nature Communications.

Response: We are glad to see the reviewer's positive comments on our revised manuscript. Thank you for your effort in reviewing it for twice. According to the further comments, we have made changes or improvements in manuscript with some more experiments. Below are the detailed replies.

Details are listed below:

1. *Page 2, lines 64-66: the authors mention three types of coke, but only specify two, soft and hard coke.*

Response: Thank you for your comment.

As discussed in Introduction, there are three types of coke formation during the MDA reaction: 1) carbidic carbon in Mo_2C , 2) molybdenum-associated coke, and 3) aromatic-type coke on acid sites. The carbidic carbon in Mo_2C and molybdenum-associated coke are regarded as soft coke while aromatic-type coke on Bronsted acid sites is considered as hard coke.

We have changed this sentence as below:

In addition, coke formation may occur during fast kinetics. The catalytic pyrolysis of CH_4 on the active $\text{Mo}_2\text{C}/\text{MoO}_x\text{C}_y$ sites leads to amorphous coke deposits (soft coke), and oligomerization and/or cracking of the intermediates (C_2H_4) and polycondensation of

formed aromatics on the Brønsted acid sites promotes polyaromatic carbonaceous deposits (hard coke)¹⁻³.

2. *Figure 1: although the reaction pathway is presented in Figure 1, I think it would be useful to have the individual elementary steps of the process proposed in the text as well.*

Response: Thanks for the reviewer's suggestion.

We have revised the manuscript to reflect the changes as below:

It is proposed that the synergistic interaction between bimetallic Pt-Bi alloy and Mo specie creates a new (A–B–D) pathway for elementary reaction on zeolite external surface, as shown in Fig. 1, which is in parallel with the conventional (A–C–D) pathway via the acidic proton-Mo-C sites in the interior surface of zeolite cages. On the external surface, the activation of the C-H bond occurs on the Mo₂C, leading to dissociative adsorption of methane to form surface alkyl intermediate (CH₃*). Instantly, the CH₃* are transferred to the surface of Pt-Bi particles for subsequential scission of C-H bonds to form CH₂* and then preferential C-C coupling of two CH₂ species instead of the deep dehydrogenation to improve the selectivity for C₂ products and the catalyst lifetime. On the other hand, the Pt-Bi alloy mitigates the mobility of the Mo species, possibly enhancing the retention of the Mo species inside the zeolite channels during reaction.

Page 3, lines 83-85: At some points of the manuscript, such as this one, it is confusing when the authors refer to which of the active species are on the external surface of the zeolite crystals or inside of the structure on the micropore surface. For instance, here, in line 83, the authors should specify they are referring to the surface within the pores.

Response: Thanks for the reviewer's suggestion.

We have carefully checked the language to avoid such confusing expression. As an example, we improved the sentences as below:

It is proposed that the synergistic interaction between bimetallic Pt-Bi alloy and Mo specie creates a new (A–B–D) pathway for elementary reaction on external surface of zeolite particle, as shown in Fig. 1, which is in parallel with the conventional (A–C–D) pathway via the acidic proton-Mo-C sites in the interior surface of zeolite cages.

4. *Page 3, lines 88-90: Although some results may be anticipated in the Introduction, I think there is no reason for referring to a particular figure (Figure S1).*

Response: We thank the reviewer for the suggestion.

Figure S1 has been moved to Results section. The figure number has been updated to S15.

5. Page 4, lines 111-112: according to the authors “Mo particle is 3-5 nm that less than pore size of ten-membered-ring (10MR) zeolite, such as MFI (ZSM-5).” This sentence should be rephrased. Moreover, a particle of 3-5 nm in size will not fit within the pores of the MFI structure, with sizes in the range of 0.5-0.55 nm. In fact, nothing can be distinguished in Figure 2-C.

Response: Thanks for the reviewer’s comment.

We agree with the reviewer that it is hard to clearly identify the size of Mo particles from the TEM or EDS results, which was limited by somehow the nature of easily cleavage of zeolite under high electron energy and the instruments. Nevertheless, some recent works have demonstrated the isolated Mo particles inside the zeolite pores^{4,5} under observation of the integrated differential phase-contrast scanning transmission electron microscopy (iDPC-STEM)⁶. Therefore, we had the discussion based on these prior results and facts.

To make our discussion more clear, we have made the changes to read as below:

“Energy-dispersive X-ray spectroscopy (EDS) result (Fig. 2D) shows the uniform elemental distribution of Mo, Pt, and Bi, where the size of Pt-Bi particle is identified to be ~30 nm, which is larger than pore size of ten-membered-ring (10MR) zeolite such as MFI (ZSM-5) with pore size of about 5.5 Å. Therefore, the Pt-Bi particles are located on the external ZSM-5 surface. In addition, it was demonstrated that the Mo-oxo complexes are inside the zeolite pores in 4 wt% Mo/ZSM-5 catalyst as examined by the integrated differential phase-contrast scanning transmission electron microscopy (iDPC-STEM)⁶.”

Literature result⁶: iDPC-STEM image of Mo/ZSM-5, showing the presence of off-center contrast in many 10-MR channels. Insets show zoomed-in areas 1, 2, and 3 respectively and corresponding intensity line profiles.

6. Page 5, line 151: Reference is made to Figures S4-S9. In these graphs, please use the same range in the scale for CH₄ conversion (0-25%).

Response: Thanks for your suggestion.

We have changed the scale for CH₄ conversion for Figures S4-S9.

7. Page 5, line 173: the authors discuss the “conversion rates”. However, the results they give are no rates but methane conversion degrees (%). This should be corrected along the whole manuscript.

Response: Thanks for the comment.

We have revised the manuscript to reflect the changes.

8. Page 6, lines 179-188: the authors discuss the reduction of coke formation when adding Pt-Bi, and relate it with “a suppressed scission of C-H bonds on the external Mo species and with the reduced mobility of acidic proton-Mo in the pore of zeolite.” Did the authors consider the possible contribution of coke precursors’ hydrogenation due to presence of Pt?

Response: Thank you for your suggestion.

We agree with reviewer that the suppression of coke formation was partially resulted from the hydrogenation of coke precursors at presence of Pt. It is well known that adding Pt to Mo/H-ZSM-5 improved the catalytic activity of Mo/H-ZSM-5 due to the suppression of carbon accumulation on the catalyst^{7,8} and promotion of the hydrogenation of carbon⁹. Our DFT results also demonstrated that in the Pt-Bi alloy, H and carbon species prefer to bind with the Pt species. The initial CH₄ activation resembles the behaviors of Mo₂C (111), and the subsequent C-H activation and C₂H₄ formation on Pt-Bi (111), which suppressed continuous scission of C-H bonds on the Mo species to format coking. In this work, the presence of bismuth further enhances the hydrogenation activity of the catalyst^{10,11}.

We have revised the manuscript to reflect the changes.

“This is probably attributed to the enhancement in the hydrogenation activity of the catalyst due to the presence of Pt-Bi and further suppressed continuous scission of C-H bonds.”

9. Figure 3A-C: please specify the TOS at which the results are compared.

Response: Thank you for your comment.

Time-on-stream (TOS) was 1.5 hours for the results. This information has been added to clarify it.

10. Page 7, lines 230-232: Please rephrase, the sentence is difficult to understand. Which previous study do the authors refer to?

Response: Thank you for the good suggestion.

We have rephrased this sentence as below:

The previous study showed that H₂ activity promotes the reducibility of Mo oxides and formation of more proximate acidic proton-Mo sites¹², leading to the preferably internal coke formation at acidic proton-Mo sites. The hydrogen treatment can remove the internal coke easily to recover the catalytic activity by hydrocracking¹³. The reaction can be described with Eq. (1).

11. Table S1: Please check the space velocity given for Mo catalyst, reference 8. Perhaps it should be 2000 and not 2.0.

Response: We thank the reviewer for the suggestion.

We have checked the space velocity in the literature and changed it in Table S1. It is 2000 mL·g_{cat}⁻¹·h⁻¹.

12. Figures 5A-C (page 9) are illustrative of the experiments performed to study the influence of the Pt-Bi alloy on the mobility of the Mo species. Are these figures (A-C) based on theoretical calculations, or only cartoons of the authors believe is happening when increasing Si/al ratio of the MFI or by adding Pt-Bi? Moreover, did the authors consider possible blocking of the pore mouths by the Pt-Bi particles as a possible cause for the reduced mobility of the Mo species?

Response: Thanks for the reviewer's comment.

For the first comment, the schematic figures in Figure 5A-C serve only the purpose of illustrating our hypothesized situations to explain why we design the different ratio of Si/Al for validating the influence of the Pt-Bi alloy on the mobility of the Mo species.

We have carefully checked the language to avoid such confusing expressions. As an example, we improved the sentences as below:

“Previous studies have demonstrated that Mo-species on the external surface predominantly form coke⁵. The Mo-carbide nanoparticles with a C/Mo ratio greater than 1.5 are more stable on an external Si site of ZSM5 than they are on an Al site, which provides the driving force for the migration of Mo-carbide from internal cages onto the external surface of the zeolite¹⁴, as highlighted by green dashed box in Fig. 5A. Therefore, the migration of active-Mo species onto the zeolite external surface is a key factor in determining the catalyst lifetime. In comparison to the conversion deactivation rate of Mo-based catalyst, the deactivation rate of Pt-Bi/Mo catalyst decreased by 15.7% (Supplementary Figure S13). We assume that this deactivation rate trend is related to the influence on the migration of active-Mo onto the ZSM-5 external surface during the reaction at the presence of the Pt-Bi alloy. Also, it is known that the amount of acidic proton-Mo sites and BASs are decreased as the increased ratio of Si/Al, which leads to the decreased benzene relative selectivity¹⁵. This influence on the mobility of the Mo species further amplifies this change of benzene relative selectivity.

To validate our hypothesis, a schematic diagram of the experiment about the different ratio of Si/Al is illustrated by Fig. 5A-C. Take the Mo/ZSM-5 catalyst as a comparison group, Fig. 5A and 5B showed some acidic proton-Mo sites and BASs were disappeared as the increased ratio of Si/Al from 25 to 140 responsible for the decreased benzene relative selectivity. For Pt-Bi/Mo/ZSM-5 catalyst, it is hypothesized that there are two possible situations (Fig. 5C). First, the Pt-Bi alloy might enhance the retention of the Mo species inside the zeolite channels during reaction, as highlighted by the red dashed box, leading to the alleviated the attenuation of C₆H₆ selectivity as the increased ratio of Si/Al. Second, the Pt-Bi alloy may promote the migration of acidic proton-Mo sites, described by the green dashed box, which promotes the attenuation of C₆H₆ selectivity.”

For the second comment, we agree with the reviewer that the Pt-Bi particles possibly block the pores. Based on our BET results in the supporting information, the addition of the Pt-Bi particles in Mo catalyst significantly affected the microporous texture properties (S_{micro} and V_{micro}) of this catalyst. This is likely due to strong interaction between Pt-Bi particles and external Bronsted acid sites on the catalyst blocking some of its pore structures¹⁶, further facilitating the retention of Mo species in the zeolite cage.

We have revised the manuscript to reflect the changes.

“As can be observed in Supplementary Table 2, the microporous texture properties (S_{micro} and V_{micro}) of Pt-Bi/Mo catalyst decreased at the addition of the Pt-Bi particles. This is likely due to strong interaction between Pt-Bi particles and external Bronsted acid sites on the catalyst blocking some of its pore structures¹⁶, further facilitating the retention of Mo species in the zeolite cage.”

13. Page 9, lines 303-304: “This result indicates the addition of the Pt promoted the reducibility of the molybdenum oxides by providing activated hydrocarbon atoms.” Could the authors explain this better?

Response: Thank you for the comment.

We have rephased this sentence as following:

“This result indicates the addition of the Pt promoted the reducibility of the molybdenum oxides to provide activated hydrocarbon atom. Because Mo oxide nanostructures reduced to carbide species (Mo_xC_y) or oxycarbide (MoO_xC_y) when CH₄ was the only reactant¹⁴. As the CH₄ molecule approached the Mo center of the Mo_xC_y cluster, the electrons of the C–H bond were partly transferred to the vacant d-orbitals of the Mo atom just above the Fermi level, subsequently leading to the C–H bond activation and the formation of activated hydrocarbon atom¹⁷. Additionally, the lower temperature makes the reduction easier¹⁸, therefore the presence of the Pt promotes the generation of Mo_xC_y for subsequent reactions.”

14. Page 10, discussion on the reaction kinetics: did the authors test the Pt-Bi catalyst (Mo-free)? I think it would be interesting to compare the activation energy for this catalyst with the other three in Figure 6C. It would also give information on the controlling step for the whole process. Which have been the reaction temperatures used for this study? How were the E_a obtained, at which TOS? I have not found this information in the Methods section.

Response: Thank you for your suggestion.

We have tested the activation energy of the Pt-Bi catalyst and compared it with other catalysts in Figure 1. The activation energy for 1%Mo/ZSM5 (66.1 kJ/mol) was very close to that of 2%Mo/ZSM5 (66.9 kJ/mol). In a clear contrast, the Pt-Bi catalyst has a larger activation energy of 103.7 kJ/mol, yet the Pt-Bi/Mo catalyst has a much smaller activation energy of 27.12 kJ/mol. This result strongly demonstrates the synergy between Pt-Bi and Mo for enhancing the reaction kinetics and verifies our purposed mechanism that the initial CH_4 activation mainly occurs on the Mo species, and the subsequent C-H activation and C_2H_4 formation ae on Pt-Bi. In addition, it is in a good agreement with the results in Fig. 3B and our DFT conclusion.

Figure 1. Activation energies for different catalysts are determined by Arrhenius relationship.

We have also added more details in the Method Section:

For the calculation of activation energy, the operating conditions were: 600 – 675 °C and 95% CH_4 . The collected data were taken on every 25 °C and at 1.5 h time-on-stream (TOS).

15. Page 12, Figure 7. Please refer to Figure S17, in the supporting information, so the reader can understand the color codes and the models used for the DFT study.

Response: Thank you for your suggestion.

We have optimized Figure 7 to make sure every detail can be read. As reviewer suggested, the color codes have been added to the title of Figure 7.

References

- 1 Song, Y. *et al.* Coke accumulation and deactivation behavior of microzeolite-based Mo/HZSM-5 in the non-oxidative methane aromatization under cyclic CH₄-H₂ feed switch mode. *Applied Catalysis A: General* **530**, 12-20 (2017).
- 2 Song, Y. *et al.* The distribution of coke formed over a multilayer Mo/HZSM-5 fixed bed in H₂ co-fed methane aromatization at 1073K: Exploration of the coking pathway. *Journal of Catalysis* **330**, 261-272, doi:<https://doi.org/10.1016/j.jcat.2015.07.017> (2015).
- 3 Song, Y., Xu, Y., Suzuki, Y., Nakagome, H. & Zhang, Z.-G. A clue to exploration of the pathway of coke formation on Mo/HZSM-5 catalyst in the non-oxidative methane dehydroaromatization at 1073K. *Applied Catalysis A: General* **482**, 387-396, doi:<https://doi.org/10.1016/j.apcata.2014.06.018> (2014).
- 4 Kosinov, N. *et al.* Stable Mo/HZSM-5 methane dehydroaromatization catalysts optimized for high-temperature calcination-regeneration. *Journal of Catalysis* **346**, 125-133, doi:<https://doi.org/10.1016/j.jcat.2016.12.006> (2017).
- 5 Kosinov, N. *et al.* Methane Dehydroaromatization by Mo/HZSM-5: Mono- or Bifunctional Catalysis? *ACS Catalysis* **7**, 520-529, doi:10.1021/acscatal.6b02497 (2017).
- 6 Wang, N. *et al.* Probing the Catalytic Active Sites of Mo/HZSM-5 and Their Deactivation during Methane Dehydroaromatization. *Cell Reports Physical Science* **2**, 100309, doi:<https://doi.org/10.1016/j.xcrp.2020.100309> (2021).
- 7 Xu, Y., Liu, S., Guo, X., Wang, L. & Xie, M. Methane activation without using oxidants over Mo/HZSM-5 zeolite catalysts. *Catalysis letters* **30**, 135-149 (1994).
- 8 Chen, L., Lin, L., Xu, Z., Zhang, T. & Li, X. Promotional effect of Pt on non-oxidative methane transformation over Mo-HZSM-5 catalyst. *Catalysis Letters* **39**, 169-172, doi:10.1007/BF00805578 (1996).
- 9 Kojima, R., Kikuchi, S., Ma, H., Bai, J. & Ichikawa, M. Promotion effects of Pt and Rh on catalytic performances of Mo/HZSM-5 and Mo/HMCM-22 in selective methane-to-benzene reaction. *Catalysis Letters* **110**, 15-21, doi:10.1007/s10562-006-0087-x (2006).
- 10 Tshabalala, T. E., Coville, N. J. & Scurrrell, M. S. Dehydroaromatization of methane over doped Pt/Mo/H-ZSM-5 zeolite catalysts: The promotional effect of tin. *Applied Catalysis A: General* **485**, 238-244, doi:<https://doi.org/10.1016/j.apcata.2014.07.022> (2014).
- 11 Zhu Chen, J. *et al.* Identification of the structure of the Bi promoted Pt non-oxidative coupling of methane catalyst: a nanoscale Pt₃Bi intermetallic alloy. *Catalysis Science & Technology* **9**, 1349-1356, doi:10.1039/C8CY02171F (2019).
- 12 Gao, W. *et al.* Dual Active Sites on Molybdenum/ZSM-5 Catalyst for Methane Dehydroaromatization: Insights from Solid-State NMR Spectroscopy.

- Angewandte Chemie International Edition* **60**, 10709-10715, doi:<https://doi.org/10.1002/anie.202017074> (2021).
- 13 Zhou, J. *et al.* Regeneration of catalysts deactivated by coke deposition: A review. *Chinese Journal of Catalysis* **41**, 1048-1061, doi:[https://doi.org/10.1016/S1872-2067\(20\)63552-5](https://doi.org/10.1016/S1872-2067(20)63552-5) (2020).
- 14 Gao, J. *et al.* Identification of molybdenum oxide nanostructures on zeolites for natural gas conversion. *Science* **348**, 686-690 (2015).
- 15 Zhao, K., Jia, L., Wang, J., Hou, B. & Li, D. The influence of the Si/Al ratio of Mo/HZSM-5 on methane non-oxidative dehydroaromatization. *New Journal of Chemistry* **43**, 4130-4136, doi:10.1039/C9NJ00114J (2019).
- 16 Mhamdi, M., Ghorbel, A. & Delahay, G. Influence of the V+Mo/Al ratio on vanadium and molybdenum speciation and catalytic properties of V–Mo–ZSM-5 prepared by solid-state reaction. *Catalysis Today* **142**, 239-244, doi:<https://doi.org/10.1016/j.cattod.2008.07.026> (2009).
- 17 Khan, T. S., Balyan, S., Mishra, S., Pant, K. K. & Haider, M. A. Mechanistic Insights into the Activity of Mo-Carbide Clusters for Methane Dehydrogenation and Carbon–Carbon Coupling Reactions To Form Ethylene in Methane Dehydroaromatization. *The Journal of Physical Chemistry C* **122**, 11754-11764, doi:10.1021/acs.jpcc.7b09275 (2018).
- 18 Auroux, A. *Calorimetry and thermal methods in catalysis*. Vol. 154 (Springer, 2013).

REVIEWERS' COMMENTS

Reviewer #3 (Remarks to the Author):

The authors have carefully addressed the comments and suggestions of the reviewers and, in my opinion, this revised manuscript is now suitable for publication in Nature Communications.

Point-to-point responses to the reviewer' comments and suggestions

Reviewer #3 (Remarks to the Author):

Comments to the Author

The authors have carefully addressed the comments and suggestions of the reviewers and, in my opinion, this revised manuscript is now suitable for publication in Nature Communications.

Reply: We thank reviewer's contributions and efforts in our manuscript. With your comments and suggestions, the paper has been significantly improved for publication.